ᵃ | **Open Peer Review** | Environmental Microbiology | Research Article

# The RND efflux system ParXY affects siderophore secretion in *Pseudomonas putida* KT2440

Nicola Victoria Stein,[1] Michelle Eder,[1] Fabienne Burr,[1] Sarah Stoss,[1] Lorenz Holzner,[2] Hans-Henning Kunz,[2] Heinrich Jung[1]

**ABSTRACT** Tripartite efflux systems transport antimicrobial agents, toxic metabolites, and siderophores from Gram-negative cells into the environment. For example, the main siderophore pyoverdine of the soil bacterium *Pseudomonas putida* KT2440 is secreted into the environment via the PvdRT-OpmQ and MdtABC-OpmB systems. Here, we looked for efflux systems that might be involved in the secretion of pyoverdine in addition to the latter two systems. Screening of different efflux systems revealed that *parX* (encoding the periplasmic adapter protein of the ParXY system) is of particular importance for bacterial growth under iron limitation. Further analysis showed that the deletion of *parX* impairs the production and secretion of pyoverdine, causing the observed growth effect. The effects were dependent on the presence of other tripartite efflux systems and the conditions of iron limitation. The results suggest that ParXY not only plays a role in antibiotic resistance, as shown previously, but also influences the secretion of siderophores in a network of overlapping activities of different tripartite efflux systems.

**IMPORTANCE** Gram-negative bacteria from the *Pseudomonas* group are survivors in various environmental niches. For example, the bacteria secrete siderophores to capture ferric ions under deficiency conditions. Tripartite efflux systems are involved in the secretion of siderophores, which are also important for antibiotic resistance. For one of these efflux systems, the resistance-nodulation-cell division transporter ParXY from the model organism *Pseudomonas putida* KT2440, we show that it influences the secretion of the siderophore pyoverdine in addition to its already known involvement in antibiotic resistance. Phenotypically, its role in pyoverdine secretion is only apparent when other pyoverdine secretion systems are inactive. The results confirm that the different tripartite efflux systems have overlapping substrate specificities and can at least partially functionally substitute for each other, especially in important physiological activities such as supplying the cell with iron ions. This fact must be taken into account when developing specific inhibitors for tripartite efflux systems.

**KEYWORDS** RND transporter, pyoverdine, ParXY, PP_3455, PP_3456, iron, siderophore, PvdRT-OpmQ, MdtABC-OpmB, MexAB, *Pseudomonas putida* KT2440, chloramphenicol

I ron is essential as a trace element for almost every organism on earth, including bacteria, fungi, plants, and animals (1). Its importance ranges from crucial processes such as bacterial development to oxygen transport in the metabolism of invertebrates (2). It is a cofactor in iron-sulfur clusters and many enzymes such as nitrogenase (3), cytochrome c, hemoglobin, or catalase (4). However, the transition from vital to toxic is reached rather quickly (1): excess iron promotes the formation of harmful reactive oxygen species through the Fenton reaction, which can destroy Fe-S clusters and cause DNA damage (4, 5).

Iron-specific transporters can take up soluble ferrous ($Fe^{2+}$) ions. However, $Fe^{2+}$ is readily oxidized under aerobic conditions to the ferric form ($Fe^{3+}$), which forms insoluble

Address correspondence to Heinrich Jung, hjung@lmu.de.

The authors declare no conflict of interest.

See the funding table on p. 15.

iron hydroxide (4). $Fe^{3+}$ can be scavenged by complexation with siderophores. Bacteria, fungi, and plants are all capable of producing these iron chelators (6), which can have different structures and molecular sizes and differ in their functional groups. Synthesis and secretion of siderophores in bacteria are regulated at the transcriptional level in response to iron availability by extracytoplasmic sigma factors, which in turn are tightly controlled by the ferric uptake regulator (Fur) (7). While strains such as *Escherichia coli* (8) and *Pseudomonas aeruginosa* (5) produce more than one siderophore, *Pseudomonas putida* KT2440 produces pyoverdine as its only siderophore. It has a high affinity for ferric ions ($k_d$ of $10^{-32}$ $M^{-1}$) (9) and can chelate ferric ions from abiotic and biotic sources (2). Pyoverdine is produced by cytoplasmic non-ribosomal peptide synthetases (NRPS) and matures in the periplasm to the final fluorescent structure (7, 10, 11). Although the mechanisms of siderophore synthesis, maturation, and uptake have been extensively studied (7), the secretion mechanisms and systems involved are only partially understood (12–15).

Gram-negative bacteria utilize ATP-binding cassettes (ABC) and resistance-nodulation-cell division (RND) efflux systems to enable the secretion of siderophores. These systems cross the inner membrane, periplasm, and outer membrane (16). Accordingly, the systems consist of proteins in the inner and outer membranes, connected by a periplasmic adaptor protein (17). These efflux systems contribute to microbial resistance to antibiotics and other toxic compounds (18–20). At the same time, they are crucial for microbial physiology, including bacterial cell communication, colonization, intracellular survival, and virulence (18, 20). The regulation of these transport systems involves a highly complex interplay of global and local transcriptional regulators and two-component systems (18, 20).

The involvement of the ABC and RND systems in the secretion of siderophores has been extensively studied in *Escherichia coli* (21). Here, AcrAB, AcrAD, and MdtABC, each together with TolC, a multifunctional outer membrane channel, form three RND systems involved in the secretion of the siderophore enterobactin (Ent) (21). In pseudomonads, the tripartite efflux system PvdRT-OpmQ (ABC type) is responsible for the secretion of newly synthesized and recycled siderophore pyoverdine. In *P. aeruginosa*, this is the only efflux system identified to date (13, 14, 22). In *P. putida* KT2440, previous studies have shown that in addition to PvdRT-OpmQ, the tripartite efflux system MdtABC-OpmB (RND type) is also involved in the secretion of the siderophore (15). First biochemical evidence for an interaction of the PvdRT-OpmQ system of *P. putida* KT2440 with pyoverdine has been recently presented, reconfirming the involvement of this system in siderophore transport (23). Inactivation of these transport systems leads to the inhibition of pyoverdine secretion and, thus, the growth of the bacterium in the presence of iron deficiency. The inhibition is partial, suggesting that at least one other system is involved in pyoverdine secretion or can take over the tasks of the deleted systems (2, 7, 15).

Here, we search for transport systems in addition to PvdRT-OpmQ and MdtABC-OpmB, which are important for the growth of *P. putida* KT2440 under iron limitation and might contribute to pyoverdine secretion. In the first experiment, genes encoding components of tripartite efflux pumps or outer membrane porins were individually deleted in a *P. putida* KT2440-derived strain, lacking functional PvdRT-OpmQ and MdtABC-OpmB ($\Delta pm$). A comparison of the growth of the resulting mutants under iron-replete and iron-deficient conditions revealed that the RND-type efflux system ParXY is crucial for growth under iron limitation. Subsequently, more detailed analyses of the role of ParXY revealed that the RND system influences growth, pyoverdine production, and secretion. The influence of ParXY depends on the presence of other efflux systems and the exact conditions of iron limitation in the culture medium, indicating a crucial role for this system in pyoverdine secretion under iron-scarce conditions.

## RESULTS AND DISCUSSION

### Screening for efflux systems and outer membrane porins critical for *P. putida* KT2440 growth under iron limitation

The tripartite efflux systems PvdRT-OpmQ and MdtABC-OpmB were previously suggested to be involved in pyoverdine secretion in *P. putida* KT2440 (15). Here, we show that *P. putida* KT2440 (wild type), *P. putida* Δ*pvdRT-opmQ*Δ*mdtA* (Δ*pm*), and *P. putida* 3E2 [no biosynthesis of pyoverdine (24)] grew under iron-rich conditions [King's B (KB) medium] with only small differences. However, under strong iron limitation [KB plus 1 mM 2'2-bipyridyl (Bip)], the growth of the Δ*pm* strain was inhibited compared to the wild type but significantly better than that of strain 3E2 (Fig. 1). The results are in agreement with our previous observations (15) and suggest that other systems besides PvdRT-OpmQ and MdtABC-OpmB may contribute to the secretion of pyoverdine.

In search of these other transport systems, genes encoding components of tripartite efflux systems [PP_0166 (*paxA*), PP_0804, PP_0906 (*mexV*), PP_1264 (*fusBCD*), PP_1516 (*mexJ*), PP_2064 (*mexH*), PP_2818 (*mexD*), PP_3302, PP_3426 (*mexF*), PP_3455 (*parX*), PP_3549 (*emrA*), PP_5173 (*triC*)] or outer membrane porins [PP_1798, PP_4519, PP_4923 (all *tolC*-like), PP_1019 (*oprB-I*), PP_1273, PP_2069, PP_2558, PP_3427 (*oprN*)] were individually deleted in the Δ*pm* strain (23). Furthermore, based on a previous report on the involvement of type 6 secretion systems (T6SSs) in pyoverdine secretion by *Pseudomonas taiwanesis* (26), we introduced the Δ*pm* mutations into a *P. putida* KT2440R-derived strain (ΔT6SS) lacking all three T6SSs (25). All resulting mutants grew about equally well under iron-rich conditions (KB medium) (Fig. 1A). However, under strong iron limitation (KB plus 1 mM Bip), a strain with the triple deletion Δ*pm*Δ*parX* (Δpp3455) showed the most striking effect besides strain 3E2 on growth compared to the Δ*pm* strain. In fact, the area under the curve (AUC) used here as a growth parameter was for strain Δ*pm*Δ*parX* about 40% of that of the Δ*pm* strain. However, in contrast to the non-producer 3E2 (2% of Δ*pm*), the Δ*pm*Δ*parX* strain showed significant growth (Fig. 1B). Minor but statistically significant effects on the growth of the Δ*pm* strain were caused by the additional deletion of PP_0166 (*paxA*), PP_3302 (RND family transporter), PP_3426, PP_3427 (inner and outer membrane components of the MexEF-OprN system), or PP_4519 and PP_4923 (*tolC*-like) (Fig. 1B). While the exact mechanism behind these effects is unknown, it should be noted that in KB plus 1 mM Bip medium, changes in the complex network of efflux systems may affect not only the circuits associated with iron acquisition (e.g., secretion of pyoverdine) but also resistance to toxic Bip. Indeed, the tripartite efflux systems TtgABC and MexEF-OprN were previously shown to be involved in the resistance to Bip (19, 27). Due to the strong iron-dependent impact of the *parX* deletion on the growth of the Δ*pm* strain, we focus below on clarifying the role of the gene (gene product) under iron limitation.

### Impact of the deletion of *parX* on chloramphenicol resistance of *P. putida* KT2440

The *parX* (PP_3455) gene encodes a periplasmic adaptor protein typically found in tripartite efflux systems of the RND type. The gene is in an operon with *parY* (PP_3456), the inner membrane component of the system. As the operon does not contain a gene encoding an outer membrane porin, *parXY* is expected to utilize the outer membrane components of other tripartite efflux systems (e.g., TtgC of TtgABC) (28). The amino acid sequences of ParX and ParY are indeed similar to the periplasmic (TtgA) and inner membrane components (TtgB) of the TtgABC system (Fig. S1). The ParXY system confers resistance to the antibiotics cefepime, aminoglycosides, and fluoroquinolones (28). Moreover, chloramphenicol treatment stimulates the expression of *parX* in *P. putida* strain KT2440R (28, 29), implicating ParXY in the transport of structurally diverse substrates.

To independently validate the loss of function of the ParXY system, we tested the effect of deletion of *parX* on chloramphenicol resistance under our conditions. We confirmed that deletion of the gene significantly increased the resistance of *P.*

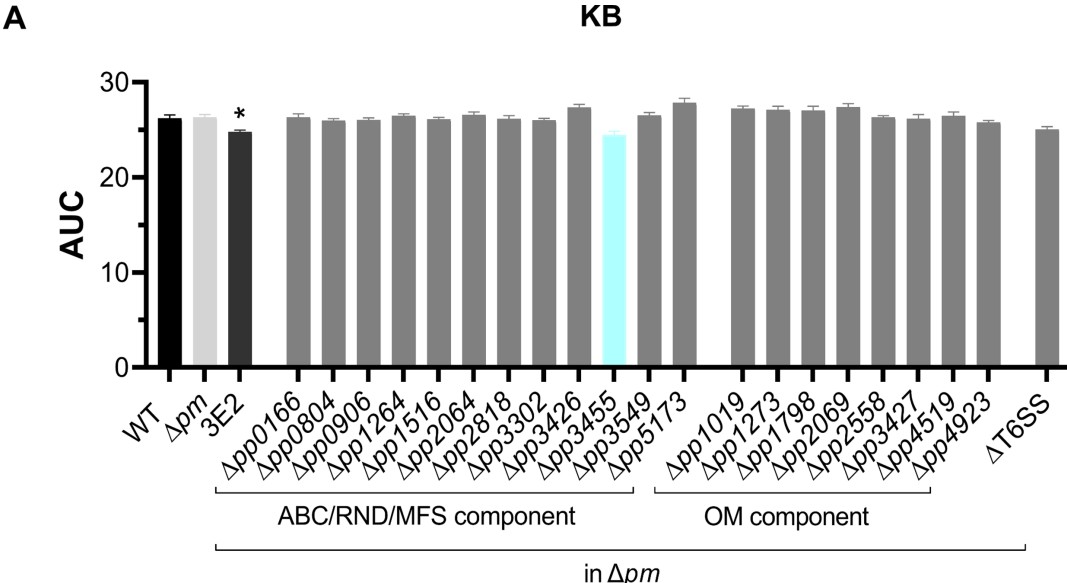

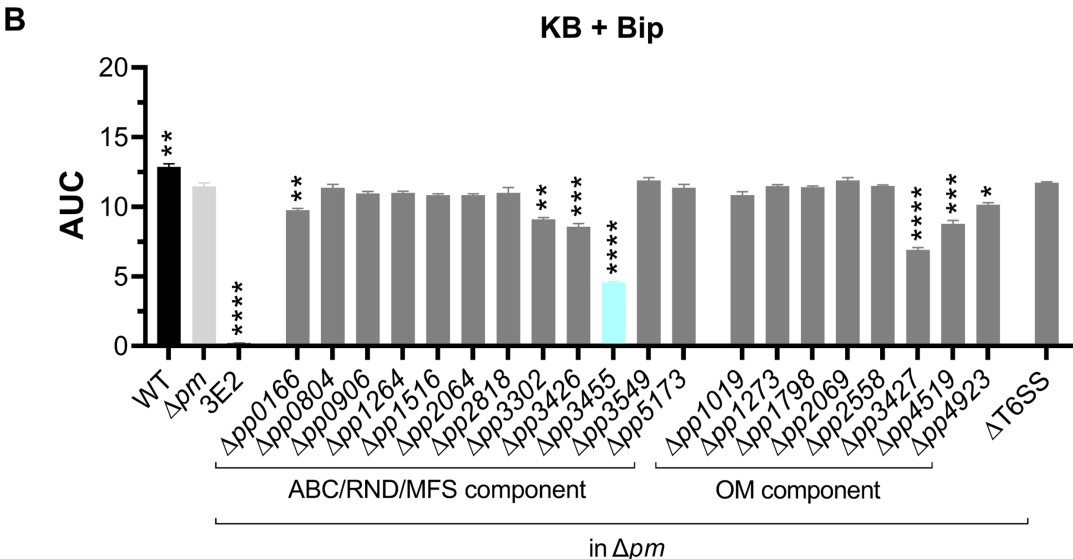

FIG 1 Influence of different tripartite efflux systems and outer membrane porins on the growth of *P. putida* KT2440 and derived mutants. (A) Cells were grown under iron-rich conditions (KB medium) and (B) strong iron limitation (KB plus 1 mM Bip). Cultures were started at an $OD_{600}$ of 0.08 and grown in a CLARIOstar Plus plate reader. For this figure, the AUC was determined using a baseline corresponding to the initial $OD_{600}$. Mutants were generated in the genetic background of the Δ*pm* strain lacking functional PvdRT-OpmQ and MdtABC-OpmB complexes. Consequently, the gene number on the x-axis indicates the third deletion introduced into the strain. In the case of ΔT6SS, we introduced the Δ*pm* deletions into a strain without functional T6SSs [Δ*tssA1*Δ*tssM2*Δ*tssM4* (25)]. For growth analysis, mutants were always grown together with the wild-type strain, the strain Δ*pm* (light gray), and the pyoverdine non-producer 3E2 (dark gray). Under iron-deplete conditions, the largest growth defects were observed for strains 3E2 and Δ*pm*Δ*parX* (Δpp3455) (cyan). Mean values of a minimum of at least four biological replicates are shown. ANOVA and Dunnett's T3 multiple comparison tests (α 0.05) were used for statistical analysis. *0.0332; **0.0021; ***<0.0002; and ****<0.0001.

*putida* KT2440 to chloramphenicol (Fig. S2). Although biochemical evidence for direct functional interactions between ParXY and TtgABC is lacking, the results support the idea that the different tripartite efflux systems in *P. putida* KT2440 and other strains not only have overlapping substrate specificities but also interact functionally.

## Impact of *parX* on colony morphology

Next, we analyzed the impact of the deletion of *parX* on the morphology of colonies (growth, color, and fluorescence) grown on KB agar plates with and without Bip. All strains (e.g., wild type, Δ*parX*, Δ*pm*, Δ*pm*Δ*parX*, pyoverdine non-producer 3E2) developed colonies of about the same size on KB agar (Fig. S3). Despite this observation, the colony of strain Δ*pm*Δ*parX* was less fluorescent than the other strains but more fluorescent than pyoverdine non-producer 3E2. On KB plus 1 mM Bip, the colony development of strain Δ*pm*Δ*parX* was poor compared to the other strains, and strain 3E2 did not grow at all (Fig. S3). For complementation, gene *parX* was cloned into pSEVA224 (30), and the colony morphology assay (CMA) was performed with cells transformed with either pSEVA224 (pSEVA) or pSEVA224-*parX* (pSEVA-*parX*) (Fig. 2). On KB agar plates, cells of strain Δ*pm*Δ*parX* transformed with the empty plasmid were less fluorescent than the colonies of strain Δ*pm* transformed with the same plasmid. Plasmid-based expression of *parX* in the Δ*pm*Δ*parX* strain restored fluorescence to levels comparable to the Δ*pm* strain. On KB plus 1 mM Bip plates, colony development of strain Δ*pm*Δ*parX* was poor, and fluorescence was low compared to the two other strains. Again, plasmid-based expression of *parX* in the Δ*pm*Δ*parX* strain restored colony development and fluorescence to levels comparable to the Δ*pm* strain. This result confirms that the deletion of *parX* in the Δ*pm* strain affects colony morphology on KB and KB plus 1 mM Bip plates and suggests a growth defect under iron limitation. On the contrary, deletion of *parX* alone in the wild type has no effect on colony morphology.

## The importance of *parX* for growth in different liquid media

For growth experiments in liquid culture, the following media were chosen for cultivation: KB (6.57 ± 0.71 µM iron), KB plus 1 mM Bip (strong iron limitation), and casamino acid (CAA) (1.45 ± 0.19 µM iron). CAA was selected because it does not contain any known toxic compound (i.e., Bip or cetyltrimethylammonium bromide) whose presence could interfere with activities of tripartite efflux and other systems, as previously shown for Bip and the TtgABC and MexEF systems (19).

The effects of *parX* deletion on growth under different iron availability conditions in liquid culture are shown in Fig. 3. In KB medium, wild type and strains Δ*parX*, Δ*pm*, and Δ*pm*Δ*parX* grew well and showed no significant differences in terms of lag phase and growth rate (Fig. 3A). In KB plus 1 mM Bip, the combined deletion of genes of the Pvd

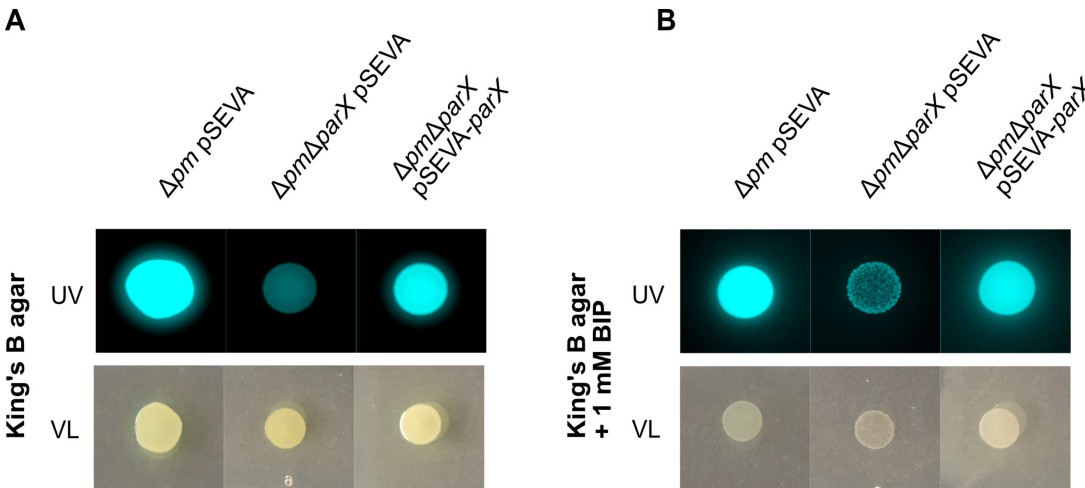

**FIG 2** Colony morphology assay for the Δ*pm*Δ*parX* strain complemented with *parX*. (A) CMA under iron-replete conditions for strains Δ*pm* transformed pSEVA224 without *parX* (control) and Δ*pm*Δ*parX* transformed with pSEVA224 or pSEVA-*parX*. (B) CMA of the strains under iron-deplete conditions. Cells were pre-grown in KB medium containing 50 µg/mL kanamycin overnight at 30°C and continuous shaking. Ten microliters of the pre-culture were spotted onto KB agar plates containing 0.5 mM IPTG (induction of *parX* expression) and 50 µg/mL kanamycin without (left) and with the addition of 1 mM Bip (right). Plates were incubated for 18 h at 30°C and imaged using a BioRad Gel Doc XR + Gel Documentation System and trans-UV. Montages were generated using Fiji (31).

and Mdt efflux systems (Δ*pm*) resulted in a prolongation of the lag phase and a reduction of the specific growth rate (Fig. 3B and D). Both effects were significantly enhanced by the additional deletion of *parX* (strain Δ*pm*Δ*parX*) (Fig. 3B and D). In contrast, individual deletion of *parX* had no significant effect on growth dynamics compared to the wild type. Growth of the 3E2 strain was severely impaired in KB plus 1 mM Bip (Fig. 3B and

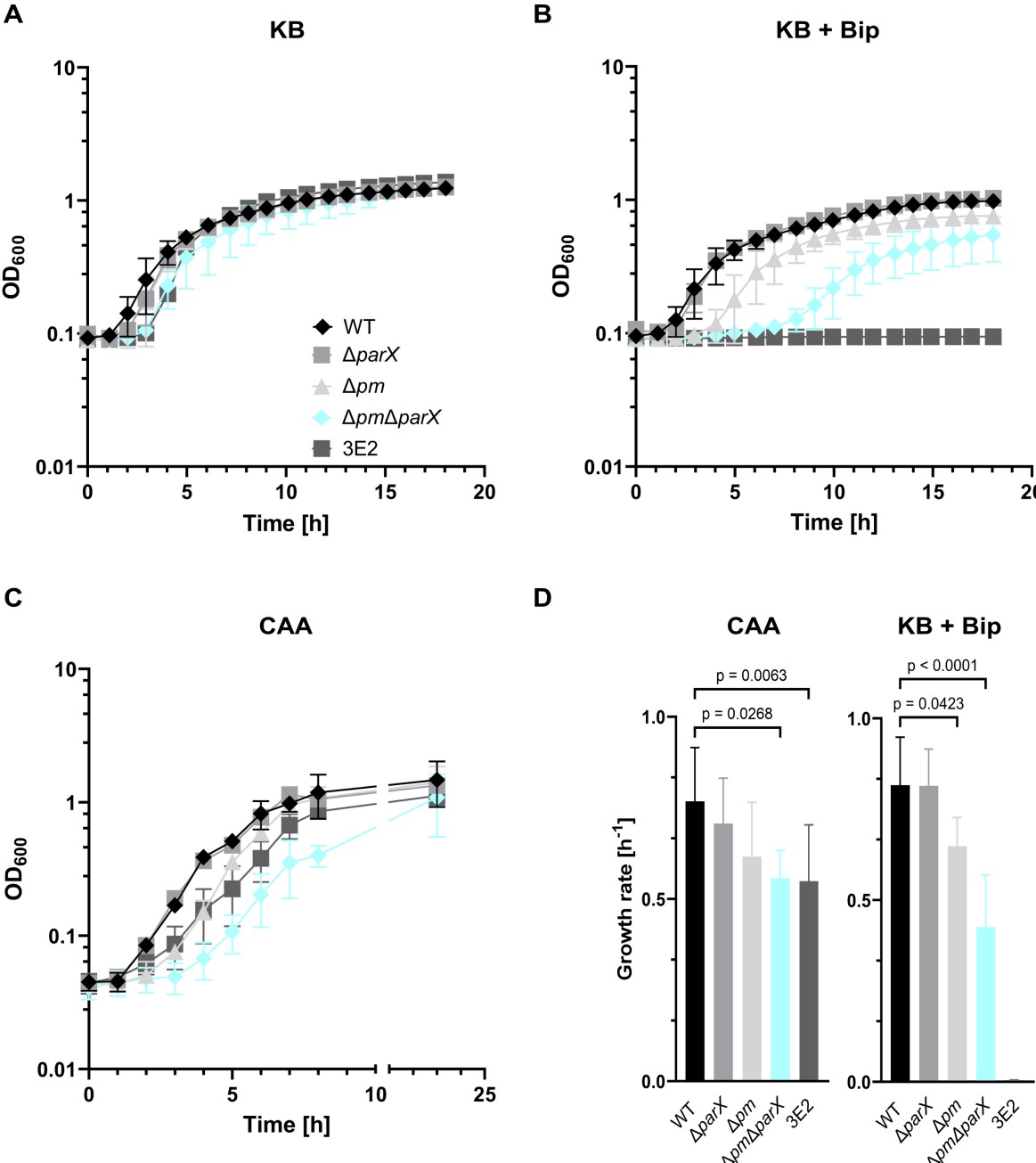

**FIG 3** Impact of *parX* deletion on the growth of *P. putida* KT2440 and derived mutants. (A) Iron-rich conditions (KB medium) and (B) strong iron limitation (KB plus 1 mM Bip) were used to grow strains *P. putida* KT2440 (WT), Δ*parX*, Δ*pm*, Δ*pm*Δ*parX*, and the pyoverdine non-producer 3E2. Bacteria were incubated with shaking for 18 h in a Tecan infinite M200 Pro plate reader. (C) For growth under weak iron limitation (CAA medium), glass flasks containing 35 mL medium were inoculated with the given strain (start OD$_{600}$ = 0.1), incubated at 30°C with constant shaking, and the cell density (OD$_{600}$) was measured hourly. (D) The specific growth rate of each strain was determined using data from the exponential phase of the growth curve under conditions of strong iron limitation (B) and weak iron limitation (C). Mean values of a minimum of three biological replicates are shown. ANOVA and Tukey's multiple comparisons tests (α 0.05) were used for statistical analysis.

D). Under weak iron limitation (CAA), the deletions had qualitatively similar, however, weaker effects on growth (Fig. 3C). Statistically, only the combined inactivation of all three efflux systems (ΔpmΔparX) resulted in significantly altered growth parameters in the CAA medium. The latter strain grew very similar to strain 3E2 in this medium (Fig. 3C and D).

For complementation, the growth of strains Δpm and ΔpmΔparX transformed with pSEVA or pSEVA-parX were analyzed in KB plus 1 mM Bip (Fig. 4A) and in the CAA medium (Fig. 4B). Under strong iron limitation, the growth of the ΔpmΔparX strain transformed with the empty plasmid was impaired (prolonged lag phase, decreased growth rate) compared to the Δpm strain containing the same plasmid, as observed above with the strains without plasmid (Fig. 3B and D). However, the growth dynamics of the ΔpmΔparX strain expressing parX from plasmid pSEVA-parX were almost identical to those of strain Δpm with pSEVA, indicating that parX expression is sufficient for complementation. Similarly, the growth of strain ΔpmΔparX with pSEVA was impaired in the CAA medium compared to strain Δpm with pSEVA. Here, the expression of parX from pSEVA-parX stimulated the growth rate of strain ΔpmΔparX to values identical to those of strain Δpm while the lag phase was still slightly prolonged (Fig. 4B; Fig. S4). The prolonged lag phase in CAA medium, which is more nutrient-poor compared to KB, could be due to a temporal and/or quantitative imbalance of parX expression by the plasmid-based system. Thereby, it is possible that an insufficient amount of complementing adapter protein ParX is produced, leading to the prolonged lag phase. Taken together, the results confirm that the efflux system ParXY is crucial for growth under iron limitation when functional PvdRT-OpmQ and MdtABC-OpmB efflux systems are absent.

## Influence of parX on the secretion of pyoverdine

Is the parX-dependent growth defect of the ΔpmΔparX strain under iron limitation related to an altered pyoverdine secretion? To answer this question, we compared the amounts of pyoverdine secreted into the culture medium (CAA) during the cultivation of the ΔpmΔparX strain with those of the wild-type, 3E2, ΔparX, and the Δpm strains.

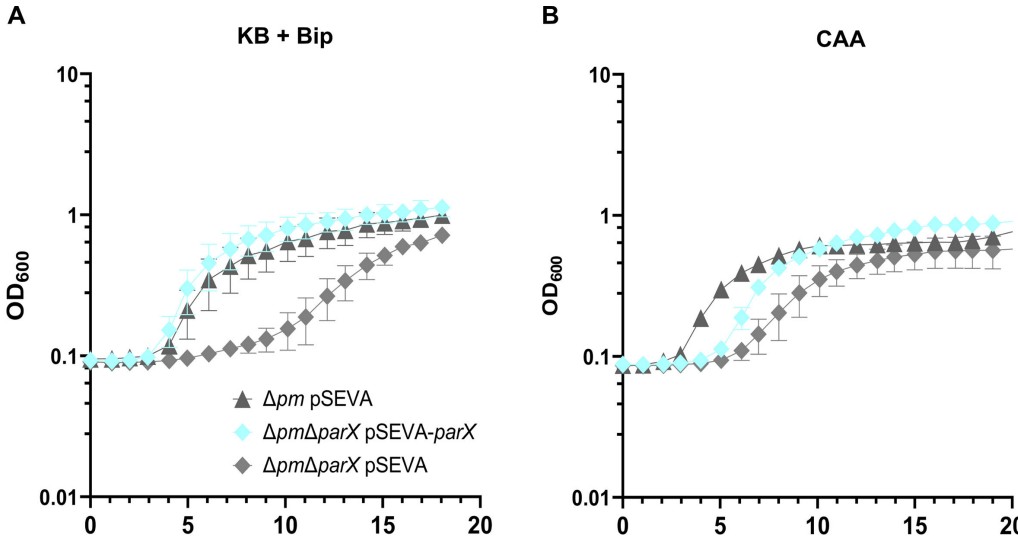

**FIG 4** Expression of parX from plasmid pSEVA224 complements the phenotype of P. putida KT2440 with the triple deletion ΔpmΔparX. (A) Strains containing the pSEVA224 plasmid encoding the adapter protein ParX (cyan) or empty plasmids (control, light, and medium gray) were grown in a 96-well plate with (A) KB medium supplemented with 1 mM 2′2-bipyridyl or (B) CAA medium. Media contained 0.5 mM IPTG for gene expression and kanamycin (50 mg/mL) for plasmid maintenance. Growth was started with an initial $OD_{600}$ of 0.1, cultures were continuously shaken, and the $OD_{600}$ was monitored for 20 h. (B) Growth rates were calculated from the exponential growth phase and are shown in Fig. S4. Mean values were calculated from a minimum of three biological replicates.

Because impaired pyoverdine secretion may be associated with the accumulation of the siderophore in the periplasm, we also determined the amounts of pyoverdine in the periplasm of the strains. In agreement with previous results (15), we found that the average amount of pyoverdine released into the CAA medium by the Δ*pm* strain (normalized to the $OD_{600}$) was significantly reduced compared with the wild type. Additional deletion of *parX* further reduced the amount of pyoverdine secreted. Deletion of *parX* alone in the wild type had no significant effect on the secretion of pyoverdine (Fig. 5A). After 24 h of growth, the difference in the amount of secreted pyoverdine between the strain Δ*pm* and Δ*pm*Δ*parX* became even more obvious (Fig. S5A). The phenotype was accompanied by a pale color of the Δ*pm*Δ*parX* culture (Fig. S5B).

The reduced amounts of pyoverdine secreted by the Δ*pm* and Δ*pm*Δ*parX* strains compared to the wild type were accompanied by increased levels of pyoverdine in the periplasm of both mutants (Fig. 5B). Remarkably, pyoverdine accumulation was slightly but significantly increased in strain Δ*pm*Δ*parX* compared to strain Δ*pm*. Deletion of *parX* alone in the wild type again had no significant effect on the amount of pyoverdine in the periplasm (Fig. 5B and C). These results suggest that an additional deletion of *parX* in the genetic background of the Δ*pm* strain further impairs pyoverdine secretion. It is unclear whether ParXY itself is capable of transporting pyoverdine or whether this is an indirect effect. In addition, in the presence of the PvdRT-OpmQ and MdtABC-OpmB systems, a secretion of pyoverdine by the ParXY system does not seem to play a role.

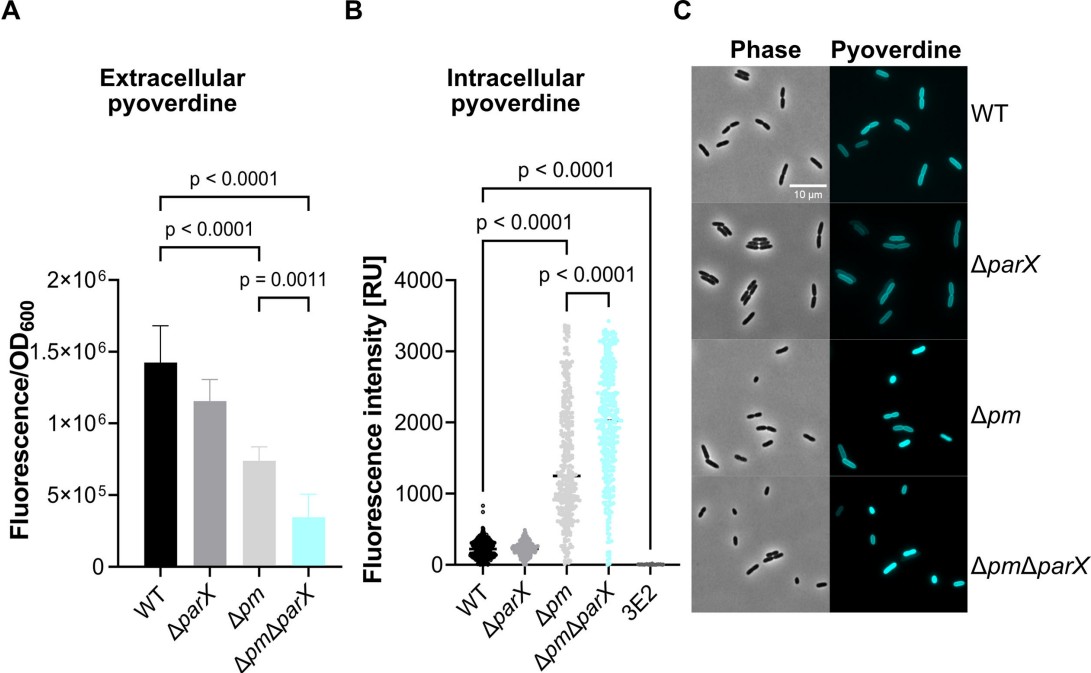

**FIG 5** Influence of the *parX* deletion on the secretion and periplasmic accumulation of pyoverdine of *P. putida* KT2440 and derived mutants. (A) Relative amounts of pyoverdine in the culture supernatants (extracellular pyoverdine). Cells were grown by shaking in CAA medium at 30°C for 7 h. One milliliter of culture was taken from each flask and centrifuged (15,700 × *g* for 3 min). Pyoverdine was determined based on its fluorescence ($\lambda_{ex}$ = 400 nm, $\lambda_{em}$ = 455 nm, Tecan infinite M200 Pro plate reader). Values were normalized to the $OD_{600}$. (B) Relative amounts of pyoverdine in the periplasm of cells (intracellular pyoverdine). Aliquots of cells grown and shaken in CAA medium at 30°C for 3 h (Fig. 3C) were analyzed on agarose pads with a Leica DMi8 inverted microscope. Pyoverdine fluorescence was observed using a Leica DFC365 FX camera (CFP channel, exposure time: 250 ms; gain: 1.5). Fiji (31) and MicrobeJ (32) tools were used to analyze the intracellular fluorescence of each individual strain for a total of 350 cells. Relative units for the corrected mean of intracellular fluorescence in the cyan channel are shown. Mean values of a minimum of three biological replicates are shown. ANOVA and Tukey's multiple comparisons tests (α 0.05) were used for statistical analysis. (C) Representative images in phase contrast and of fluorescence microscopy of WT, Δ*parX*, Δ*pm*, and Δ*pm*Δ*parX*.

## Effect of externally added FeCl$_3$, CuSO$_4$, and pyoverdine on growth of strain Δ*pm*Δ*parX*

To further test whether *parX* plays a role in iron acquisition via pyoverdine, the effect of externally added FeCl$_3$ (Fig. 6A) and pyoverdine (Fig. 6B) on the growth of the Δ*pm*Δ*parX* strain was analyzed in comparison to the wild-type and the Δ*pm* strains. Since pyoverdines are also known to bind Cu$^{2+}$, CuSO$_4$ was included in the analysis. Growth experiments were performed in CAA medium in a 96-well microtiter plate format. The addition of 1 µM FeCl$_3$ (Fig. 6A) stimulated the growth of the Δ*pm*Δ*parX* strain to levels comparable to that of the Δ*pm* strain. In contrast, the addition of 1 µM CuSO$_4$ did not significantly increase the growth of the mutants (Fig. 6B). Supplementation with 10 µM pyoverdine yielded the best growth for the Δ*pm*Δ*parX* strain (Fig. 6B). Since the addition of iron or pyoverdine stimulates the growth of the Δ*pm*Δ*parX* strain, the defects caused by the mutations (including the *parX* deletion) are likely to be due to disturbances in pyoverdine-dependent iron uptake. Similar to the effects shown in CAA medium, the growth defects of mutants from Fig. 1 and 3 in KB supplemented with 1 mM Bip could also be complemented by the addition of FeCl$_3$ at concentrations close to the Bip complex (one Fe$^{3+}$ interacts with three Bip) to KB 1 mM Bip medium (Fig. S6). In agreement with the results in CAA medium, these data support the idea that the observed *parX* effects are not due to changes in Bip resistance but to iron supply.

## Dependence of *parX* expression on iron availability

If ParXY indeed plays a role in siderophore-mediated iron uptake, the question arises whether the expression of the corresponding operon is dependent on iron availability. To answer this question, the promoter region of the *parXY* operon was fused to the *lux* gene cluster in plasmid pBBR1-MCS5-*lux* (28). Subsequently, the luminescence of *P. putida* KT2440 transformed with the resulting plasmid was analyzed (Fig. 7). Based on the high luminescence values measured compared to the negative control (reporter plasmid lacking the *parXY* promoter region), the *parXY* operon was induced in all media used. However, there were significant differences between the luminescence values that correlated with iron availability. For example, the addition of 10 µM FeCl$_3$ to the

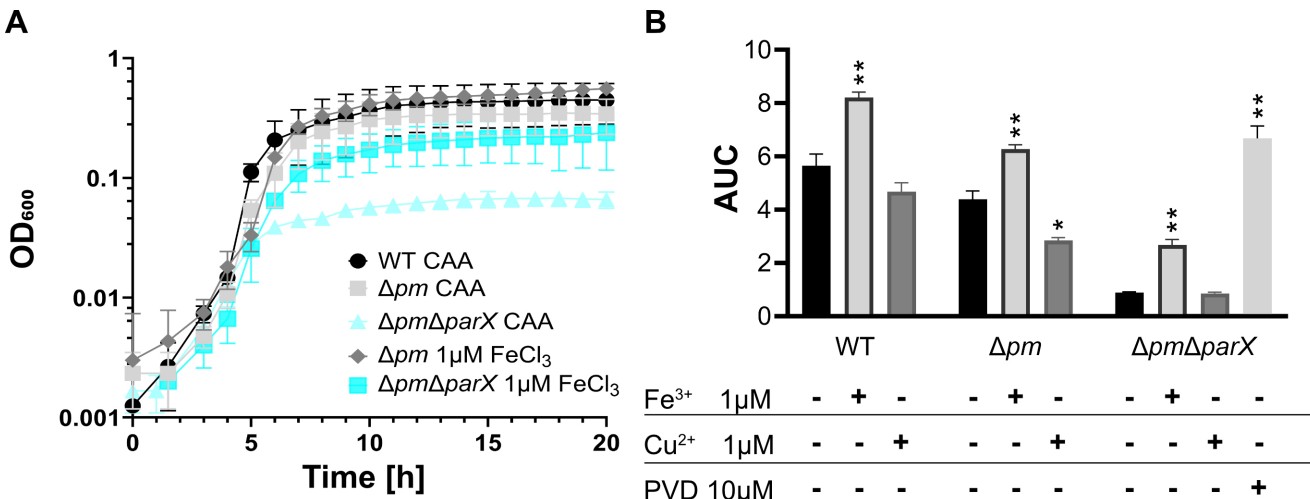

**FIG 6** Rescue of Δ*parX* phenotype mediated by the addition of external FeCl$_3$ and pyoverdine. (A) WT *P. putida* KT2440, Δ*pm*, and triple deletion Δ*pm*Δ*parX* were grown in CAA medium for 20 h. For deletion mutants Δ*pm* and triple deletion Δ*pm*Δ*parX*, the medium was supplemented with 1 µM of FeCl$_3$. (B) Additionally, WT *P. putida* KT2440, Δ*pm*, and triple deletion Δ*pm*Δ*parX* were grown in CAA medium without (black) and with the addition of either 1 µM FeCl$_3$ (medium gray), 1 µM CuSO$_4$ (dark gray), or 10 µM pyoverdine (light gray). The AUC for all tested conditions was determined using the mean initial OD$_{600}$ of each curve as a baseline: cells were grown for 20 h with continuous shaking with an orbital amplitude of 2, and optical density was measured in a Tecan infinite M200 Pro plate reader. Mean values for a minimum of three biological replicates are shown. ANOVA and Dunnett's T3 multiple comparisons tests (α 0.05) were used for statistical analysis, always comparing against CAA medium. *0.0332 and **0.0021.

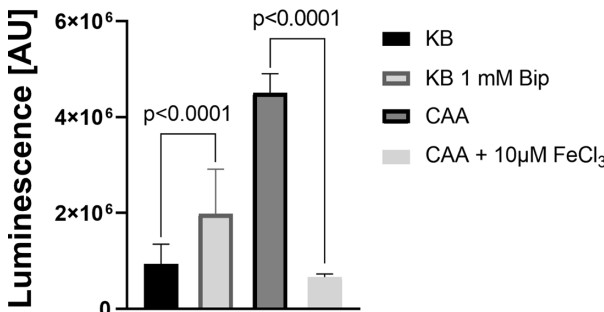

**FIG 7** Impact of iron availability on *parXY* expression in *P. putida* KT2440. Expression was analyzed based on a transcriptional fusion of the promoter region of *parXY* with the *luxCDABE* operon in plasmid pBBR1 (34). Cells of *P. putida* KT2440 were cultivated in KB, KB plus 1 mM Bip, CAA, or CAA medium supplemented with 10 μM $FeCl_3$, and luminescence was detected when the $OD_{600}$ reached a value of 0.4 (exponential growth phase). Mean values of at least three biological replicates are shown. Mean values for respective measurements with the empty vector are $4.5 \times 10^4$ (KB), $1.0 \times 10^5$ (KB 1 mM Bip), $1.0 \times 10^5$ (CAA), and $6.6 \times 10^4$ (CAA 10 μM $FeCl_3$). ANOVA and Tukey's multiple comparisons tests (α 0.05) were used for statistical analysis of KB vs KB 1 mM Bip or CAA vs CAA + 10 μM $FeCl_3$, respectively.

CAA medium resulted in an approximately sevenfold reduction in luminescence. The addition of 1 mM Bip to the KB medium resulted in a twofold increase in luminescence compared to KB alone. The results suggest that iron limitation stimulates the expression of the *parXY* operon. The mechanism behind this stimulation is not known. Unlike the promoters of *pvdRT-opmQ* and *mdtABC-opmB*, the *parXY* operon was not predicted to have a binding site for the Fur-dependent sigma factor PvdS (33). Instead, *parXY* expression was thought to be controlled by a two-component system (encoded by genes PP_3453 and PP_3454) immediately upstream of the operon (28). The stimulus perceived by the two-component system is not known. Based on sequence similarities with the RstBA system of *E. coli,* it has been speculated that PP_3453/PP_3454 interacts with another two-component system similar to PhoPQ, which senses fluctuations in extracellular $Mg^{2+}$ concentration (28). Clearly, further studies are needed to understand the role of PP_3453/PP_3454.

## Effect of the deletion of genes encoding components of PvdRT-OpmQ, MdtABC-OpmB, and ParXY on gene expression

It is well known that tripartite efflux systems have overlapping substrate specificities and can functionally substitute for each other (18, 27, 35). In this context, we hypothesized that the loss of one or more efflux systems involved in pyoverdine secretion leads to the stimulation of the expression of genes of alternative efflux systems as compensation. To determine whether the PvdRT-OpmQ, MdtABC-OpmB, and ParXY efflux systems affect each other on the one hand and pyoverdine synthesis in terms of gene expression on the other, the promoters of the operons of these systems and of the pyoverdine synthesis gene *pvdL* (encodes a non-ribosomal peptide synthetase) were individually fused to the *lux* gene cluster in pBBR1-MCS5-*lux* (34). Subsequent analysis of gene expression by luminescence measurements in wild type and mutants with single or combined gene deletions revealed no or only relatively minor effects (Fig. 8). For example, although the results from above showed that the ParXY system became physiologically important when PvdRT-OpmQ and MdtABC-OpmB were not functional, the individual or combined deletion of genes of the latter two systems had only very little effect on the expression of *parXY* (Fig. 8A). Only inactivation of the MdtABC-OpmB system by gene deletion stimulated expression of PvdRT-OpmQ system genes and vice versa (Fig. 8A), confirming the above hypothesis. Since expression of pyoverdine-related genes is controlled by sigma factors (e.g., PvdS), which in turn are regulated by Fur (15, 36, 37), a reduced secretion of pyoverdine and, as a consequence, a reduced uptake of iron into cells may

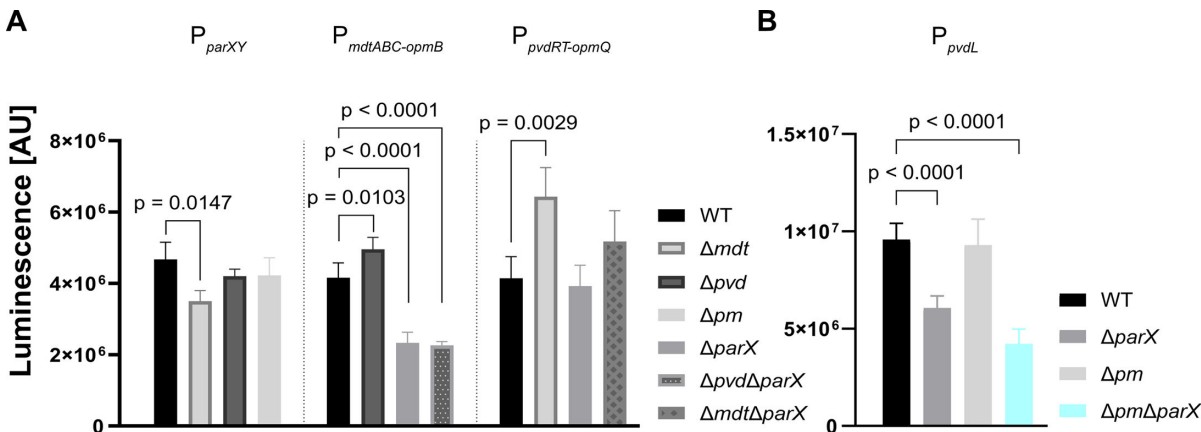

**FIG 8** Influence of the deletion of *parX*, *mdtA*, and *pvdRT-opmQ* on the expression of genes encoding tripartite efflux systems and a pyoverdine synthesis gene. Cells of *P. putida* KT2440 (WT) and the given derived mutants were transformed with pBBR1 (34), containing transcriptional fusions of the promoter region of the investigated efflux systems or the NRPS *pvdL* with the *luxCDABE* operon. Cells were cultivated in CAA medium, and the influence of the deletion of individual efflux systems (A) on the activities of the promoter of *parXY* ($P_{parXY}$), *mdtABC-opmB* ($P_{mdtABC-opmB}$), and *pvdRT-opmQ* ($P_{pvdRT-opmQ}$) and (B) on *pvdL* ($P_{pvdL}$) was tested. The graphs represent the absolute data (AU) of luciferase activity at an optical density of $OD_{600} = 0.4$ (exponential phase). Mean values of at least three biological replicates are shown. ANOVA and Dunnett's multiple comparisons tests (α 0.05) with the WT as a control strain were used for statistical analysis.

affect Fur signaling and stimulate expression of the remaining functional efflux system. Contrary to the promoters of *pvdRT-opmQ* and *mdtABC-opmB*, the *parXY* operon was not predicted to have a PvdS binding site (33).

Surprisingly, deletion of *parX* resulted in approximately twofold reduced expression of the *mdtABC-opmB* operon (Fig. 8A). Similarly, expression of *pvdL* was approximately twofold reduced upon *parX* deletion, while deletion of the two other efflux systems (strain Δ*pm*) had no impact on *pvdL* expression (Fig. 8B). The physiological significance of the latter phenomena is unclear. For the Δ*pm*Δ*parX* mutant, one could speculate that reduced pyoverdine production in the absence of three (putative) pyoverdine secretion systems prevents the accumulation of pyoverdine in the periplasm at toxic concentrations. The accumulation of intracellular siderophores can indeed have drastic effects on cell viability (38).

## Conclusion

This study showed that the ParXY system is crucial for growth under conditions of iron limitation when the pyoverdine-secreting tripartite systems PvdRT-OpmQ and MdtABC-OpmB are inactive. The growth defect is associated with decreased secretion of pyoverdine into the environment and accumulation of the siderophore in the periplasm. Growth of a mutant in which all three efflux systems were inactivated by gene deletion was restored to wild type levels by the addition of ferric iron or pyoverdine. In addition, expression of the *parXY* operon in the wild type was stimulated by iron limitation via an as yet unknown mechanism. Overall, the results suggest that ParXY contributes to the secretion of pyoverdine in addition to PvdRT-OpmQ and MdtABC-OpmB. Considering the strength of the phenotypes of mutants with individual and combined gene deletions of the three systems [this publication and reference (15)], PvdRT-OpmQ seems to be the main pyoverdine secretion system of *P. putida* KT2440, followed by MdtABC-OpmB. Since the deletion of *parX* alone does not have a significant effect on growth under iron limitation as well as pyoverdine secretion, it remains questionable whether the proposed ability of ParXY to secrete pyoverdine in the presence of the other two tripartite efflux systems is of physiological significance. However, if required, the ParXY system might be able to contribute to the secretion of siderophores with the help of TtgC or another outer membrane porin (Fig. 9). Nevertheless, to unequivocally demonstrate that the ParXY system is able to contribute to the secretion of the siderophore pyoverdine, biochemical

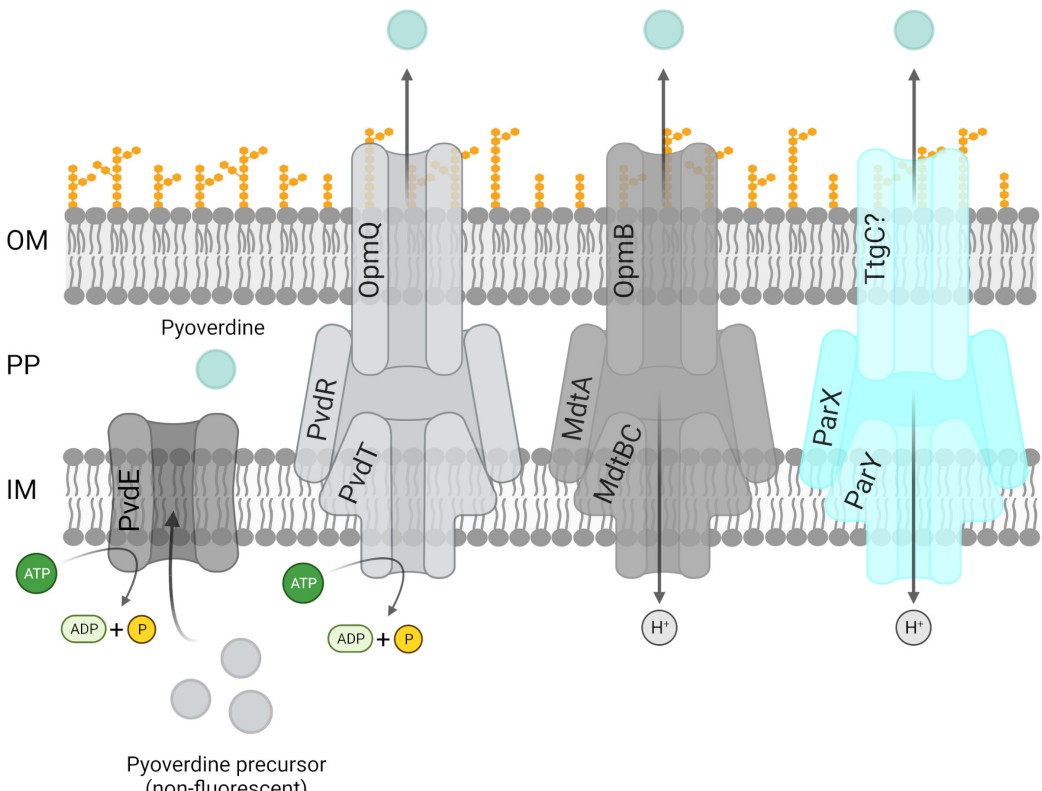

**FIG 9** Proposed model for efflux pumps involved in pyoverdine secretion in *P. putida* KT2440. Pyoverdine precursor is synthesized in the cytosol and transported into the periplasmic space (PP) via the ABC transporter PvdE. Several efflux systems are proposed to be involved in this process: PvdRT-OpmQ, MdtABC-OpmB (15), and ParXY-OMP. Initial biochemical evidence for an interaction of PvdRT-OpmQ with pyoverdine has recently been published (23).

evidence of direct binding and transport by the system is required, as has been provided in initial experiments for the PvdRT-OpmQ system (23). Finally, the increased resistance of the ΔparX strain to chloramphenicol suggests functional interactions of ParXY with other RND efflux systems (e.g., TtgABC). Taken together, the results are consistent with the idea that individual or combined deletion of tripartite efflux systems can have major effects on the physiological state of bacteria. This was particularly evident in a recent analysis in which six RND systems were simultaneously removed in *P. aeruginosa* (17). The resulting cells were characterized by iron starvation and modified lipid A.

## MATERIALS AND METHODS

### Bacterial strains and cultivation

*Escherichia coli* and *P. putida* strains and plasmids used for this study are listed in Tables S1 and S2. *Pseudomonas* strains were cultivated in casamino acid medium containing 0.5% casein hydrolysate, 6.77 mM $K_2HPO_4$, and 1.02 mM $MgSO_4 x H_2O$ under aerobic conditions by constant shaking at 180 rpm at 30°C (24). For colony morphology analysis, overnight cultures of cells were grown in King's B medium (39) containing 2% peptone, 0.15% $K_2HPO_4$, 0.15% $MgSO_4 x (7H_2O)$, and 1% (wt/vol) glycerol. Cells transformed with plasmids were grown in the presence of 50 µg/mL kanamycin (pSEVA224, pNPTS138-R6KT) or 30 µg/mL gentamycin (pBBR1-MCS5-*lux*). For measurements of growth and luminescence in the 96-well format, the gentamycin concentration was reduced to 15 µg/mL. Plasmid-based gene expression was induced with 0.5 mM isopropyl β-d-1-thi-ogalactopyranoside (IPTG). For storage, cells were grown in lysogeny broth LB medium (1% tryptone/peptone, 1% NaCl, 0.5% yeast extract) overnight, supplemented with

glycerol freezing medium (48.75% glycerol, 12 mM KCl, 12 mM NaCl, 0.3 mM $MgSO_4$), frozen in liquid nitrogen, and kept at $-80°C$. For agar plates, culture media were supplemented with 1.5% agar and poured into Petri dishes. In addition, *P. putida* KT2440 and derived strains were grown on Cetrimide agar plates (40). After inoculation, colonies developed at 30°C overnight. The resulting plates were kept at 4°C.

## Generation of plasmids and mutants

Oligonucleotides used during this study are listed in Table S3. For the generation of deletion mutants listed in Table S1, homologous recombination with the pNPTS138-R6KT system (41) was used, as described by Henríquez et al. (15). For complementation, the gene PP_3455 (*parX*) was amplified by PCR and cloned into the multiple cloning site (MCS) of pSEVA224 (30). For gene expression analyses, putative promoter regions were determined using the BPROM tool (42). Identified regions were cloned into pBBR1-MCS5-*luxCDABE* (34). PCR and sequencing were used to confirm the final plasmids and strains.

## Growth analyses

For the initial growth analysis in Fig. 1, cells from overnight cultures in KB medium were used to inoculate KB medium with and without the addition of 1 mM 2′2-bipyridyl in a 96-well plate (200 µL). Growth was monitored using a CLARIOstar Plus (BMG LABTECH) with hourly measurements of the optical density at 600 nm ($OD_{600}$). *P. putida* KT2440 (WT) served as a control strain for each setup. The AUC was determined using GraphPad Prism version 9.4.1 and a minimum of four biological replicates.

For further growth analysis in CAA medium in flasks, *P. putida* strains were initially grown in LB medium for 8 h. The resulting cells were used to start an overnight culture in CAA medium. This step helps the cells to adapt to iron limitation. Centrifugation ($15,700 \times g$, 3 min) was used to obtain the cells from CAA cultures. Cells were washed and used to inoculate 35 mL of fresh CAA medium (initial $OD_{600} = 0.05$). Growth curves were recorded under continuous shaking at 180 rpm and 30°C, and the $OD_{600}$ was measured. Complementation, recovery experiments with $FeCl_3$, $CuSO_4$, or pyoverdine, and chloramphenicol susceptibility testing were performed in a 96-well plate (Greiner) using a Tecan Infinite M200 Pro plate reader or a CLARIOstar Plus (BMG LABTECH) without initial washing. The growth rate was calculated from the exponential phase of the growth curve. For growth recovery experiments using purified pyoverdine, the siderophore was extracted following a modified protocol based on Meyer et al. (43), as recently described in Stein et al. (23). All recovery experiments were performed in CAA and KB medium. For CAA, $FeCl_3$ and $CuSO_4$ were added at a final concentration of 1 µM, whereas pyoverdine was supplemented at 10 µM prior to measurements. For KB, the medium contained 1 mM 2′2-bipyridyl and 100, 200, 300, or 400 µM of FeCl3. Susceptibility testing towards chloramphenicol was performed in Mueller Hinton (MH) medium (Sigma-Aldrich). Here, pre-cultures in MH medium were used to inoculate MH medium containing 0, 40, 80, 100, 120 to 140 µg/mL chloramphenicol (96-well plates, 200 µL medium per well, initial $OD_{600} = 0.1$, orbital amplitude of 2).

## Colony morphology assay

With some modifications, the experiment was conducted according to the protocol of Sakhtah et al. (44). Here, 10 µL of overnight cultures in KB medium was spotted onto KB plates with and without 1 mM of the iron chelator 2′2-bipyridyl and incubated at 30°C for 18 h. Agar plates were supplemented with 0.5 mM IPTG and 50 µg/mL of kanamycin for complementation assays. After incubation, the colonies were photographed under natural and UV light (BioRad Gel Doc XR + Gel Documentation System). Final images were generated using Fiji (31).

## Determination of pyoverdine in the supernatant

*Pseudomonas putida* strains were grown in CAA medium for 7 h. A 1 mL aliquot of the culture was centrifuged (15,700 × *g*, 3 min). The fluorescence of the resulting supernatant was determined with Tecan infinite M200 Pro plate reader using an excitation wavelength $\lambda_{ex}$ of 400 nm and an emission wavelength $\lambda_{em}$ of 455 nm. Values were normalized by the $OD_{600}$ of the original culture.

## Total reflection X-ray fluorescence

To analyze the element composition, total X-ray fluorescence was performed on a Bruker T-Star following a protocol described by Höhner et al. (45). For iron determination in bacterial media, the protocol was adapted as follows: media were mixed with an internal standard solution in a 1:1 ratio to a final concentration of 1 mg/mL gallium, 50 mg/mL scandium, and 0.2% polyvinyl alcohol. Subsequently, samples were dried on quartz-glass carriers (Bruker). Iron concentrations were determined via total X-ray fluorescence with Mo-K excitation at 50 kV using gallium as the standard element.

## Single-cell fluorescence analysis

*Pseudomonas putida* strains were grown in CAA medium for 3 h, harvested, and washed by centrifugation (15,700 × *g*, 3 min). The resulting cells were spotted on agar pads (2% agarose), and imaging was performed with a Leica DMi8 inverted microscope equipped with a Leica DFC365 FX camera (Wetzlar, Germany). Phase contrast and cyan filter with a $\lambda_{ex}$ of 436 nm and a $\lambda_{em}$ of 480 nm with a 40 nm bandwidth and an exposure of 250 ms, gain 4, and 100% intensity were used to capture the images. Fluorescence quantification and image analysis were done using Fiji (31) and MicrobeJ (32) tools.

## Luciferase activity assay

Overnight cultures (5 mL LB medium with 30 µg/mL gentamycin, incubated at 30°C) of *P. putida* transformed with plasmids pBBR1-P*parXY*-*lux*, pBBR1-P*mdtABC-opmB*-*lux*, pBBR1-P*pvdRT-opmQ*-*lux*, pBBR1-P*PP_4243*-*lux* (PvdL), or pBBR1-MCS5-*lux* (control) were used to inoculate CAA medium containing 15 µg/mL gentamycin (96-well plate format, 200 µL medium per well, initial $OD_{600}$ = 0.1). Plates were incubated in a CLARIOstar Plus (BMG LABTECH) plate reader at 30°C for 20 h and double orbital shaking with 600 rpm. After subtracting the background signal, the luminescence values measured at an $OD_{600}$ of 0.4 (exponential phase) were plotted individually for *P. putida* strains containing either control or plasmids with indicated promoter regions.

## Statistical analysis and figure visualization

GraphPad Prism version 9.4.1 for Windows (GraphPad Software, San Diego, California, USA) was used to plot and statistically analyze the data. Ordinary one-way ANOVA, Tukey's multiple comparison test, Dunnett's (T3) multiple comparisons test, Kruskal-Wallis test, two-way ANOVA, and Šídák's multiple comparisons tests were used as indicated. The generation of figures was done using Affinity Designer version 1.10.4.1198 for Windows.

### ACKNOWLEDGMENTS

We thank Tania Henríquez for the generation of strains *P. putida* KT2440 Δ*pm* and Δ*mdt*. The proposed model was generated using Biorender.com.

Research and the APC in the group of H.J. were supported by the Deutsche Forschungsgemeinschaft, projects JU333/6-1, and the Faculty of Biology, LMU Munich. The funder had no role in the study design, data collection, and interpretation or the decision to submit the work for publication.

*P. putida* strain 3E2 and *P. putida* strain KT2440R Δ*tssA1*Δ*tssM2*Δ*tssM3* (ΔT6SS) were kindly provided by Pierre Cornelis (Vrije Universiteit Brussels, Belgium) and Alain Filloux (Imperial College London), respectively.

N.V.S. and H.J. planned and supervised the experiments; N.V.S., M.E., F.B., and S.S. generated the strains and plasmids and performed growth curves; N.V.S. and S.S. conducted pyoverdine measurements; N.V.S. performed microscopy and image evaluation; N.V.S. and F.B. performed the luciferase assays; N.V.S., M.E., and F.B. performed the susceptibility tests and rescue experiments; L.H. and H.-H.K. performed TXRF measurements; N.V.S. and H.J. wrote the manuscript; H.-H.K. and H.J. supervised and acquired funding. All authors reviewed and approved the manuscript.

## AUTHOR AFFILIATIONS

[1]Microbiology, Faculty of Biology, Ludwig Maximilian University Munich, Martinsried, Germany
[2]Plant Biochemistry and Physiology, Faculty of Biology, Ludwig Maximilian University Munich, Martinsried, Germany

## PRESENT ADDRESS

Sarah Stoss, Technical University Munich, Munich, Germany

## AUTHOR ORCIDs

Nicola Victoria Stein http://orcid.org/0000-0002-2641-0628
Michelle Eder http://orcid.org/0000-0002-3548-520X
Fabienne Burr http://orcid.org/0009-0005-9593-7828
Lorenz Holzner http://orcid.org/0000-0002-1938-2307
Hans-Henning Kunz http://orcid.org/0000-0001-8000-0817
Heinrich Jung http://orcid.org/0000-0002-5450-3063

## FUNDING

| Funder | Grant(s) | Author(s) |
| --- | --- | --- |
| Deutsche Forschungsgemeinschaft (DFG) | JU333/6-1 | Heinrich Jung |
| Ludwig-Maximilians-Universität München (LMU) | Faculty of Biology | Heinrich Jung |

## AUTHOR CONTRIBUTIONS

Nicola Victoria Stein, Conceptualization, Data curation, Formal analysis, Investigation, Methodology, Supervision, Validation, Visualization, Writing – original draft, Writing – review and editing | Michelle Eder, Data curation, Investigation, Writing – review and editing | Fabienne Burr, Formal analysis, Investigation, Writing – review and editing | Sarah Stoss, Formal analysis, Investigation, Writing – review and editing | Lorenz Holzner, Formal analysis, Methodology, Writing – review and editing | Hans-Henning Kunz, Formal analysis, Methodology, Supervision, Writing – review and editing | Heinrich Jung, Conceptualization, Formal analysis, Funding acquisition, Methodology, Project administration, Resources, Software, Supervision, Validation, Writing – original draft, Writing – review and editing

## DATA AVAILABILITY

Supporting data, additional oligonucleotide sequences, and findings of this study are available from the corresponding authors for provable reasons.

## ADDITIONAL FILES

The following material is available online.

## Supplemental Material

**Supplemental material (Spectrum02300-23-s0001.pdf).** Tables S1 to S3 and Fig. S1 to S6.

## Open Peer Review

**PEER REVIEW HISTORY (review-history.pdf).** An accounting of the reviewer comments and feedback.

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
