## [Reviewer comments · Microbiology Spectrum]

Microbiology Spectrum

The RND efflux system ParXY affects pyoverdine secretion in *Pseudomonas putida* KT2440

Nicola Stein, Michelle Eder, Fabienne Burr, Sarah Stoss, Lorenz Holzner, Hans-Henning Kunz, and Heinrich Jung

Corresponding Author(s): Heinrich Jung, Ludwig Maximilian University of Munich

Review Timeline:

Submission Date:	June 2, 2023
Editorial Decision:	July 6, 2023
Revision Received:	August 23, 2023
Accepted:	August 28, 2023

Editor: Giordano Rampioni

Reviewer(s): The reviewers have opted to remain anonymous.

Transaction Report:

DOI: <https://doi.org/10.1128/spectrum.02300-23>

July 6, 2023

Prof. Heinrich Jung
Ludwig Maximilian University of Munich
Planegg-Martinsried
Germany

Re: Spectrum02300-23 (Involvement of the RND efflux system ParXY in pyoverdine secretion in *Pseudomonas putida* KT2440)

Dear Prof. Heinrich Jung:

Thank you for submitting your manuscript to Microbiology Spectrum. Your manuscript has been evaluated by two Reviewers with expertise in the area addressed in your study and it was the consensus view of these Reviewers that your paper contains interesting data with significant potential impact. However, both Reviewers recommended substantial modifications before manuscript acceptance. In particular, both Reviewers recommend to include additional control experiments to support the role played by the ParXY efflux system in pyoverdine secretion. Overall, I will be glad to consider for publication in Microbiology Spectrum a revised version of your manuscript addressing all the comments raised by the Reviewers. Please consider that the revised manuscript could undergo a second full review as the modifications needed are significant.

Link Not Available

Sincerely,

Giordano Rampioni

Journals Department
Reviewer comments:

Reviewer #1 (Comments for the Author):

The authors show that deletion of the tripartite efflux pump ParXY in *Pseudomonas putida*, together with deletion of two other pump systems, PvdRT-OpmQ and MdtABC-OpmB, generates a growth defect in Fe-limited conditions compared to the wild-type strain and the double mutant, and also that complementation of these strains with the parX gene on a plasmid, as well as media supplementation with FeCl₃ or pyoverdine, restores this growth phenotype. They also show that the triple mutant generally

produces less pyoverdine than the wild-type and the double mutant. Therefore, it is possible to conclude that the increased growth defects observed in the triple mutant under iron-limiting conditions may be caused by reduced pyoverdine production. This research is relevant under the viewpoint of bacterial physiology, helping to understand the role of ParXY in pyoverdine production and survival under iron-limiting conditions. Moreover, this work is also important from the clinical viewpoint, since the use of efflux pump inhibitors is one of the antimicrobial strategies currently under development, precisely because these efflux systems usually play a role in important physiological processes beyond their contributions to antibiotic resistance. The authors further conclude that the ParXY efflux system is involved in the extrusion of pyoverdine. However, I think that one important aspect should be addressed to corroborate this conclusion:

- Figure 5 indicates that mutation of the *parX* gene in a genetic background lacking the PvdRT-OpmQ and MdtABC-OpmB systems results in both a lower accumulation of pyoverdine in the supernatant and a slightly higher intracellular accumulation compared to the wild-type and double mutant strains. Based on these results, the authors conclude that ParXY is able to extrude pyoverdine. However, both experiments were performed at different points of cell growth: 7 hours (supernatant) and 3 hours post-inoculation (intracellular). Considering that pyoverdine production and extrusion is a very finely regulated process, extracellular and intracellular pyoverdine accumulation assays should be carried out in cultures presenting the same physiological conditions of growth and cell density for further comparison. Therefore, I recommend the authors to repeat these experiments using the same incubation time for both samples and, if possible, using the same biological sample.

This work has analyzed multiple growth conditions in different strains and culture media, obtaining results that support most of the conclusions presented. However, they use different methodologies to analyze and to represent bacterial growth, which makes it difficult to compare and assimilate the results. Some examples are:

- Figures 3, 4 and 6: the authors measured the growth kinetics over 18-24 hours. However, only figures 3 and 4 are represented as a growth curve. In addition, figures 1 and 6, which represent specific points of the growth cycle rather than growth kinetics, use different incubation times (figure 1: 5 hours; figure 6: 10 hours). I think that, whenever possible, bacterial growth should be analyzed at different times, and when a single numerical parameter would be used to represent the bacterial growth, you should use a more representative parameter such as doubling time, maximum optical density, or area under the curve (this last with cultures incubated for the same time).

- Furthermore, it is surprising that the authors did not use in this work any minimal medium such as M9, which can be supplemented with universal carbon sources such as succinate or citrate. This would have allowed to the authors a much better control of iron concentrations through exogenous FeCl₃ supplementation (see reference doi: 10.1111/1462-2920.14263). I suggest the authors include it in this or other subsequent work that may be related to pyoverdine production or any other limiting growth condition.

Finally, I think that the methodology used to demonstrate that the expression of the *parX* gene is influenced by iron availability is lacking of control test. When using this type of reporter genes (*lux*, GFP, etc.) to measure the expression of a gene, it is necessary to include a control construction in the study strains, similarly to that used for *parX* complementation assay in which an empty pSEVA was included. However, I have not found these controls in the paper or in the methodology. In order to confirm that the observed light variations are only observed when the *lux* operon is under the transcriptional control of the P*parX* promoter, I suggest the authors include in the experiments strains with constitutive expression of *luxCDABE* operon (DOI: 10.1128/AAC.01095-19).

Line 1-2: The title should be more in line with the evidences provided by the authors. Although the authors demonstrate that lower production and extracellular accumulation of pyoverdine was associated with loss-of-function of ParXY, the possible expulsion of pyoverdine through this efflux systems remain unclear. Therefore, I suggest changing the title or addressing the suggestions proposed below.

Line 35-36: I suggest naming the two main efflux systems involved in pyoverdine expulsion, PvdRT-OpmQ and MdtABC-OpmB.

Line 83-84: The authors mention that "The efflux systems are crucial for microbial physiology" but no more information about that are given. It could be interesting to add some more information in this section.

Line 147: It is not clear if *OprN* is PP_3427 (as mentioned in line 136) or PP_2558 (as here mentioned). Please clarify.

Line 146-148: Was a statistical test applied to affirm this? If so, this should be indicated in the figure or in the text.

Line 191-193: This is not the standard methodology to show growth defects. I suggest that the conclusion of this experiment should focus on the morphology of the colony. However, growth defects could be suggested.

Line 194-195: I am unable to appreciate such differences.

Line 199-200: I am unable to appreciate such differences. In my opinion, only large differences in fluorescence or yellow pigment should be compared in this type of experiment

Line 207-2012: *Pseudomonas putida*, like other *Pseudomonas* species, produces many pigments and fluorescent compounds. Therefore, it should not be assumed that all fluorescence emission (not wavelength specific) and yellow colour is produced by pyoverdine. Moreover, a more intense yellow halo with lower fluorescence was observed in some colonies of the figure S2 (i.e.: colonies 4 and 6). Therefore, I consider that these experiments are not determinant to clarify the role of the ParXY system in pyoverdine production.

Line 228 (figure S4): The caption and the flask identifications seem to be in error. The flask shown as " Δ *parX*" is labelled in the image as "3E2", which is a different strain. In addition, the correlation between the number and the strain identification used in this image is not the same than used in figure S2.

Line 262-264: As mentioned above, I suggest that the authors base their conclusions on more specific techniques for the detection of pyoverdine than yellow intensity and fluorescence under ultraviolet light, some of which have already been employed in this work.

Line 265-270: Could the authors clarify why different incubation times were chosen to measure extracellular (7 hours) and intracellular (3 hours) pyoverdine accumulation? This makes it difficult to compare the two experiments, since cell density and

physiological state of the population could be different, which in turns could be affecting pyoverdine production and accumulation. I recommend repeating both measurements using one of the two times (3 or 7 hours), and, if possible, using the same biological sample. Otherwise, I don't think it can be concluded that the ParXY efflux system plays a role in pyoverdine extrusion.

Line 281-283: Why is a single point (10 hours) represented when growth kinetics has been done for 17 hours? There are better numerical parameters that can be used to compare bacterial growth kinetics, such as the area under the growth curve, the maximum OD, or the doubling time.

Line 297-298: The addition of BIP causes the largest growth defects in the triple mutant, which the authors associate with the lower availability of iron in the medium. However, CAA medium induces the best expression of the parX gene, which the authors also associate with the low availability of iron in the medium. This discrepancy leaves doubts about the toxicity of BIP, which could be contributing to the growth defects observed in the triple mutant independently of the iron availability. To solve this, the authors could compare the growth of the different mutants in KB + BIP supplemented with FeCL3, which should restore growth in the double and triple mutant.

Line 340-341: I suggest to the authors to move this section after the section "Screening for efflux systems and outer membrane porins critical for *P. putida* KT2440 growth under iron-limitation", in order to validate the loss-of-function of the different efflux systems.

Line 353: Iron instead of Ion?

Line 381: Cetrимide agar medium or plates have not been included in this section.

Line 444: I recommend filtering the supernatant through a 0.22 um pore-size membrane to remove any cell debris.

Line 636-637: I suggest that the authors carry out 3 or more biological replicates of each strain.

Reviewer #2 (Comments for the Author):

The authors describe the contribution of the ParXY efflux pump to pyoverdine secretion in *Pseudomonas putida* KT2440. The manuscript is difficult to read due to several repetitions. A revision of the manuscript to limit itself to essential information would improve the manuscript.

- My main comment concerns the media used. We didn't know the iron concentrations in the different media used (CCA, KB, KB +1 mM 2'2-bipyridyl, cetrимide). It would be preferable to indicate the iron concentration of the media used. I don't think it's enough to indicate weak, strong, rich, given the major role played by iron. You have also used 1 mM of 2'2-bipyridyl (Bip) to deplete the medium of metal ions. But this compound is not specific for iron. Bip is capable of binding Cr, Fe, Co, Ru and Rh. How can we ensure that the observed effect is not linked to the chelation of other metal ions?

- The title mentions ParXY, but Only the parX gene has been deleted. What is the impact of deleting parY gene or both on the phenotype?

- The parXY efflux pump works with TtgC, is the phenotype persist with ttgC deletion mutant. it would be interesting to know if the ParXY efflux pump binds TtgC in this context or another outer membrane protein.

-Lines 117 to 123. The sentences correspond to a repetition of the previous section (introduction). The authors should focus on the new data.

- The section on colony morphology is long. Only the essential messages should be provided. The paragraph needs to be revised and made more concise. Furthermore, all data are subjective and require quantitative values, colony size, fluorescence need to be measured and quantified. ParX deletion and complementation have a similar impact in iron-rich and iron-poor environments (Figure 2). How do you explain this?

- section "importance of parX for growth in liquid culture".

Lines 214 to 217. Results already presented on lines 140 to 145. Figures 3A and 3B correspond to the data shown in Figures 1 A and 1 B. How can you explain the difference in results for the 5-hour reading between the two figures?

- Lines 214 to 230: There is no new data compared with the results already presented.

- How do you explain the different results between the agar medium (impact of ParXY in the presence of high and low iron concentrations) and the liquid medium (impact of ParXY only in the low iron concentration)?

- Line 244. You suggest that parX complementation with the low copy number is responsible for the partial restoration of growth in CAA. But this is not the case in KB + BIP, and the same plasmid is used.

- Line 289. This is not very clear. According to figure 7, parX expression already seems to be activated in iron-rich medium (at a basal level?). CAA medium, which contains more iron than KB + 1 mM BIP medium, seems to induce parX gene expression to a greater extent. Does this not agree with your hypothesis that parX is induced under low iron concentrations? The results should be checked by RT-qPCR. In addition, the expression level of the two-component system downstream of the parX gene should also be determined.

- Supplementary Table S4 and Figures S1, S2, S3, S4, S5, S6 and S7 are not useful.

Staff Comments:

Preparing Revision Guidelines

Please return the manuscript within 60 days; if you cannot complete the modification within this time period, please contact me. If you do not wish to modify the manuscript and prefer to submit it to another journal, please notify me of your decision immediately so that the manuscript may be formally withdrawn from consideration by Microbiology Spectrum.

Letter of response

Dear Dr. Rampioni,

We would like to thank you and the reviewers for the valuable feedback on our manuscript entitled "Involvement of the RND efflux system ParXY in pyoverdine secretion in *Pseudomonas putida* KT2440". We have carefully reviewed each comment and revised it accordingly. Changes are highlighted in yellow in the Word file of the revised manuscript. Below, we provide clarifications and information on all changes and additions in the manuscript.

Reviewer #1:

Query 1

Figure 5 indicates that mutation of the *parX* gene in a genetic background lacking the PvdRT-OpmQ and MdtABC-OpmB systems results in both a lower accumulation of pyoverdine in the supernatant and a slightly higher intracellular accumulation compared to the wild-type and double mutant strains. Based on these results, the authors conclude that ParXY is able to extrude pyoverdine. However, both experiments were performed at different points of cell growth: 7 hours (supernatant) and 3 hours post-inoculation (intracellular). Considering that pyoverdine production and extrusion is a very finely regulated process, extracellular and intracellular pyoverdine accumulation assays should be carried out in cultures presenting the same physiological conditions of growth and cell density for further comparison. Therefore, I recommend the authors to repeat these experiments using the same incubation time for both samples and, if possible, using the same biological sample.

Answer: We appreciate the reviewer's suggestion to perform extracellular and intracellular pyoverdine accumulation assays under the same physiological conditions of growth and cell density. We repeated these experiments using the same incubation time for both samples and the same biological sample to facilitate accurate comparison. The new results are presented in the revised Fig. 5. The new results are consistent with the statements in the original manuscript. Accordingly, the additional deletion of *parX* from a *P. putida* strain lacking the efflux pumps PvdRT-OpmQ and MdtABC-OpmQ results in further accumulation of pyoverdine in the periplasm and reduced secretion of the siderophore into the culture medium.

Query 2

This work has analyzed multiple growth conditions in different strains and culture media, obtaining results that support most of the conclusions presented. However, they use different methodologies to analyze and represent bacterial growth, which makes it difficult to compare and assimilate the results. Some examples are:

- Figures 3, 4 and 6: the authors measured the growth kinetics over 18-24 hours. However, only figures 3 and 4 are represented as a growth curve. In addition, figures 1 and 6, which represent specific points of the growth cycle rather than growth kinetics, use different incubation times (figure 1: 5 hours; figure 6: 10 hours). I think that, whenever possible, bacterial growth should be analyzed at different times, and when a single numerical parameter would be used to represent the bacterial growth, you should use a more representative parameter such as doubling time, maximum optical density, or area under the curve (this last with cultures incubated for the same time).

Answer: All growth data in this manuscript are based on optical density measurements over time (18-24 h). For clarity, only optical density at one time point was shown in the original Figure 6. Based on the reviewer's comments, the area under the curve (AUC) was determined for identical periods of incubation for the revised versions of Figure 6. The legend was adapted accordingly.

We also have complete growth curves for the growth experiments shown in Figure 1 with 25 strains under each of the two growth conditions. Of course, we will provide comprehensive growth data upon request. Since this is an initial screening and the growth details of the strains that are important for the conclusions of the manuscript are shown in Figures 3, 4, and 6, we would like to refrain from presenting all growth curves. However, we follow the reviewer's suggestion and replace the OD₆₀₀ values after 5 hours of growth with the "area under the curve (AUC)." The differences between strains now reflect the sum of changes in lag phase duration, growth rate, and maximum growth yield. Also, in this modified graph, the deletion of *parX* shows the greatest effect on growth under iron limitation (KB +Bip). In addition to this important observation, other effects (e.g., reduced OD₆₀₀ after 5 hours of growth in KB medium as a result of deletion of the genes of the T6SSs) are not visible in the AUC-based graph. Figure 1 and the text have been modified accordingly (1st paragraph of the Results and Discussion section). We believe the graphs represent a now much more coherent representation of the data we generated and would like to thank the reviewer for the suggestion of including the parameter AUC in the manuscript.

Query 3

Furthermore, it is surprising that the authors did not use in this work any minimal medium such as M9, which can be supplemented with universal carbon sources such as succinate or citrate. This would have allowed to the authors a much better control of iron concentrations through exogenous FeCl₃ supplementation (see reference doi: 10.1111/1462-2920.14263). I suggest the authors include it in this or other subsequent work that may be related to pyoverdine production or any other limiting growth condition.

Answer: We agree with the reviewer that with a defined minimal medium such as M9 with succinate as a C-source, growth conditions can be better controlled than with complex media. Indeed, in many experiments, we use M9 medium with an appropriate C source. However, here, we chose to use CAA medium. The reasons are: (i) In experiments using M9 medium, cell growth in 96-well plate format is poor and sometimes difficult to reproduce even when using the C source succinate, which is preferred by pseudomonads if the cells contain one or two plasmids stabilized by addition of antibiotics for complementation purposes or for expression analysis by reporter gene fusions. When CAA medium (or other complex media) is used, the requirements, especially for anabolic activities, are lower, and the cells are less stressed when cultured in the presence of antibiotics to maintain plasmids. Growth in CAA medium is also well reproducible in the 96 well plate format under antibiotic stress conditions. (ii) The CAA medium we used has an iron concentration of 1.45 μM. The M9 minimal medium we prepared without adding iron has an iron concentration of 2.34 μM. In terms of iron content, our M9 medium, therefore, offers no advantage compared to CAA medium. In the reference given by the reviewer, an iron concentration of 0.3 μM is given for M9 medium. This is indeed interesting. In future experiments requiring iron-limiting conditions, we will try to lower the iron content in our M9 medium toward the published value by using more deionized water, controlling the quality of the salts used, and using appropriate vessels (e.g., more consequently plastic instead of glass).

Query 4

Finally, I think that the methodology used to demonstrate that the expression of the *parX* gene is influenced by iron availability is lacking of control test. When using this type of reporter genes (*lux*, GFP, etc.) to measure the expression of a gene, it is necessary to include a control construction in the study strains, similarly to that used for *parX* complementation assay in which an empty pSEVA was included. However, I have not found these controls in the paper or in the methodology. In order to confirm that the observed light variations are only observed when the *lux* operon is under the transcriptional control of the P*parX* promoter, I suggest the authors include in the experiments strains with constitutive expression of *luxCDABE* operon (DOI: 10.1128/AAC.01095-19).

Answer: To ensure that the measured luciferase activities (Figures 7 and 8) from cells with plasmids pBBR1-P_{parXY}-lux, pBBR1-P_{mdtABC-opmB}-lux, pBBR1-P_{pvdRT-opmQ}-lux, and pBBR1-P_{pvdL}-lux were indeed from the indicated promoters, we used plasmid pBBR1-lux (without promoter) as a control. Luminescence values measured at an optical density of 0.4 ranged from 9 to 15% of the values for the plasmid containing a respective promoter sequence. We included the values for the empty plasmids in the legend of Figure 7. Since the same reporter system was used for Figure 8 and we do not expect any difference, the respective values were not added for each individual strain in Figure 8. We also added this information to the *Materials and Methods, Luciferase activity assay*.

Query 5

Line 1-2: The title should be more in line with the evidences provided by the authors. Although the authors demonstrate that lower production and extracellular accumulation of pyoverdine was associated with loss-of-function of ParXY, the possible expulsion of pyoverdine through this efflux systems remain unclear. Therefore, I suggest changing the title or addressing the suggestions proposed below.

Answer: At the suggestion of the reviewers, we have changed the title of the manuscript. It states now (lines 1-2):

“The RND efflux system ParXY affects siderophore secretion in *Pseudomonas putida* KT2440”

Query 6

Line 35-36: I suggest naming the two main efflux systems involved in pyoverdine expulsion, PvdRT-OpmQ and MdtABC-OpmB.

Answer: The two main efflux systems involved in pyoverdine secretion are named PvdRT-OpmQ and MdtABC-OpmB.

Query 7

Line 83-84: The authors mention that "The efflux systems are crucial for microbial physiology" but no more information about that are given. It could be interesting to add some more information in this section.

Answer: We added the information requested by the reviewer as follows (lines 83-84):

“At the same time, they are crucial for microbial physiology, including bacterial cell communication, colonization, intracellular survival, and virulence [18, 20].

Query 8

Line 147: It is not clear if OprN is PP_3427 (as mentioned in line 136) or PP_2558 (as here mentioned). Please clarify.

Answer: Thank you for pointing this out. The correct gene number for OprN is PP_3427. We have corrected the number in line 138.

Query 9

Line 146-148: Was a statistical test applied to affirm this? If so, this should be indicated in the figure or in the text.

Answer: ANOVA and Tukey's or Dunnett's T3 multiple comparison tests (α 0.05) were used for statistical analysis. This information was added to the legends of Figures 1, 3, and 6. The *p* values were added to the Figure 1 and 6.

Query 10

Line 191-193: This is not the standard methodology to show growth defects. I suggest that the conclusion of this experiment should focus on the morphology of the colony. However, growth defects could be suggested.

Answer: Following the remark of the reviewer, we revised our statement. Now we state (lines 197-200):

“This result confirms that the deletion of *parX* in the Δpm strain affects colony morphology on KB and KB plus 1 mM Bip plates and suggests a growth defect under iron limitation. On the contrary, deletion of *parX* alone in the wild type has no effect on colony morphology.”

Query 11

Line 194-195: I am unable to appreciate such differences.

Answer: The statement was deleted, and the paragraph was rewritten and shortened. Extending Figure 2, the original Figure S2 (now Figure S3) shows now colony formation on KB and KB + 1 mM bip plates only.

Query 12

Line 199-200: I am unable to appreciate such differences. In my opinion, only large differences in fluorescence or yellow pigment should be compared in this type of experiment

Answer: The statement was deleted, and the paragraph was rewritten and shortened. Extending Figure 2, Figure S3 shows now colony formation on KB and KB + 1 mM Bip plates only. The original extended version of Figure S3 was deleted.

Query 13

Line 207-2012: *Pseudomonas putida*, like other *Pseudomonas* species, produces many pigments and fluorescent compounds. Therefore, it should not be assumed that all fluorescence emission (not wavelength specific) and yellow colour is produced by pyoverdine. Moreover, a more intense yellow halo with lower fluorescence was observed in some colonies of the figure S2 (i.e.: colonies 4 and 6). Therefore, I consider that these experiments are not determinant to clarify the role of the ParXY system in pyoverdine production.

Answer: *Pseudomonas putida* KT2440 produces various fluorescent compounds like many other cells. However, to our knowledge, pyoverdine is the only fluorescent compound that accumulates in the periplasm and is secreted into the medium in larger amounts. To avoid confusion with possible other fluorescent compounds, we used pyoverdine non-producer 3E2 as a negative control. And the strain is not fluorescent under our conditions. Separately, we revised the entire chapter on „The ParXY system and its impact on colony morphology“. The following changes were made:

- The first paragraph, which contains already known information about ParXY (lines 161 to 171 in the original manuscript, now 160 - 179), has been moved to the beginning of the paragraph on chloramphenicol sensitivity as part of our response to Query 19.
- As a consequence of the deletion of the results of the colony morphology assay on different media (see above), the second paragraph listing the strains and characterizing the different media used for the experiments (lines 172 to 184 in the original manuscript, now 180 - 200) was shortened. Part of the content moved to the paragraph on the „Importance of *parX* for growth in different liquid media“ (lines 202 – 239).

- The remaining paragraphs of the chapter (lines 185 to 212) have been shortened and now refer only to Fig. 2 of the manuscript and the revised Fig. S2. The title of the paragraph has been changed to "Impact of parX on colony morphology."

Query 14

Line 228 (figure S4): The caption and the flask identifications seem to be in error. The flask shown as " Δ parX" is labeled in the image as "3E2", which is a different strain. In addition, the correlation between the number and the strain identification used in this image is not the same as that used in Figure S2.

Answer: We rechecked the identity of the colonies and found that reviewer 1 was right and our labeling was wrong. Due to query 15 of reviewer 2, the maximum amount of figures in the Supplementary Information file was reduced, wherefore we also removed this figure in the revised version. We nevertheless appreciate the very accurate and precise comment of reviewer 1 at this point.

Query 15

Line 262-264: As mentioned above, I suggest that the authors base their conclusions on more specific techniques for the detection of pyoverdine than yellow intensity and fluorescence under ultraviolet light, some of which have already been employed in this work.

Answer: In agreement with the changes described above, the statement was deleted, and the respective Figures were reduced to only showing Figure S5 and time point 24h for the color of the cultures.

Query 16

Line 265-270: Could the authors clarify why different incubation times were chosen to measure extracellular (7 hours) and intracellular (3 hours) pyoverdine accumulation? This makes it difficult to compare the two experiments since cell density and physiological state of the population could be different, which in turn could be affecting pyoverdine production and accumulation. I recommend repeating both measurements using one of the two times (3 or 7 hours), and, if possible, using the same biological sample. Otherwise, I don't think it can be concluded that the ParXY efflux system plays a role in pyoverdine extrusion.

Answer: We repeated the experiments. Extracellular and intracellular pyoverdine was analyzed after 7 h of growth. Fig. 5 was revised accordingly. Despite some quantitative changes, the original conclusions were confirmed.

Query 17

Line 281-283: Why is a single point (10 hours) represented when growth kinetics has been done for 17 hours? There are better numerical parameters that can be used to compare bacterial growth kinetics, such as the area under the growth curve, the maximum OD, or the doubling time.

Answer: The data provided here refer to Figure 6, which we revised to show example growth curves for the effect of Fe^{3+} on growth, including strains WT, Δpm , and $\Delta pm \Delta parX$. In addition, we have replaced the OD_{600} after 10 h of growth with the area under the curve (AUC), as suggested by the reviewer (see also our answer to Query 2 above). The text was adapted accordingly.

Query 18

Line 297-298: The addition of BIP causes the largest growth defects in the triple mutant, which the authors associate with the lower availability of iron in the medium. However, CAA medium induces the best expression of the parX gene, which the authors also associate with the low availability of iron in the medium. This discrepancy leaves doubts about the toxicity of BIP,

which could be contributing to the growth defects observed in the triple mutant independently of the iron availability. To solve this, the authors could compare the growth of the different mutants in KB + BIP supplemented with FeCl₃, which should restore growth in the double and triple mutant.

Answer: We followed the reviewer's advice and supplemented KB + Bip with FeCl₃. We added an additional Figure to the Supplementary Information file (Fig. S6).

Query 19

Line 340-341: I suggest to the authors to move this section after the section "Screening for efflux systems and outer membrane porins critical for *P. putida* KT2440 growth under iron-limitation", in order to validate the loss-of-function of the different efflux systems.

Answer: On the advice of the reviewer, we have included the section "Effects of deletion of *parX* on chloramphenicol resistance of *P. putida* KT2440" immediately after the section "Screening for efflux systems and outer membrane porins critical for *P. putida* KT2440 growth under iron-limitation". The text was adapted as highlighted in the manuscript.

Query 20

Line 353: Iron instead of Ion?

Answer: We corrected the mistake. Thank you.

Query 21

Line 381: Cetrimide agar medium or plates have not been included in this section.

Answer: We added the missing information to the respective paragraph in *Material and Methods, Bacterial Strains and cultivation* as follows:

"In addition, *P. putida* KT2440 and derived strains were grown on Cetrimide agar plates (Brown, V.I., Lowbury, E.J.L. 1965). (Lines 383 – 384)

Query 22

Line 444: I recommend filtering the supernatant through a 0.22 µm pore-size membrane to remove any cell debris.

Answer: Yes, filtration is a good way to get spent media cell-free. We decided to separate the cells by centrifugation. We remove only the top half of the supernatant. The results are very reproducible, and the method allows a high sample throughput.

Query 23

Line 636-637: I suggest that the authors carry out 3 or more biological replicates of each strain.

Answer: The results shown in Figure 1 are based on up to 10 biological replicates. For two strains, there were only two biological replicates. As we revised the manuscript, we repeated the experiments. Now, there are at least four biological replicates per strain. Figure 1 was revised accordingly, showing the area under the curve and statistical analysis.

Reviewer #2:

Query 1

The authors describe the contribution of the ParXY efflux pump to pyoverdine secretion in *Pseudomonas putida* KT2440. The manuscript is difficult to read due to several repetitions. A revision of the manuscript to limit itself to essential information would improve the manuscript.

Answer: The manuscript has been extensively revised and shortened. Some of the chapters have been reorganized (see also our responses to Reviewer 1). We hope that the manuscript is now easier to read.

Query 2

My main comment concerns the media used. We didn't know the iron concentrations in the different media used (CCA, KB, KB +1 mM 2'2-bipyridyl, cetrimide). It would be preferable to indicate the iron concentration of the media used. I don't think it's enough to indicate weak, strong, rich, given the major role played by iron. You have also used 1 mM of 2'2-bipyridyl (Bip) to deplete the medium of metal ions. But this compound is not specific for iron. Bip is capable of binding Cr, Fe, Co, Ru and Rh. How can we ensure that the observed effect is not linked to the chelation of other metal ions?

Answer: We determined the iron concentrations in the media by total reflection X-ray fluorescence (TXRF). The iron concentrations are: CAA 1.45 μ M, KB 6.57 μ M, KB plus 1 mM Bip, same as KB; only Fe³⁺ is expected to be bound to Bip. The iron concentrations of the respective media have been included in the section "*The importance of parX for growth in liquid culture*". In order to avoid interference by toxic Bip, we used Bip only mainly for the initial screening (Figs. 1 and 2). Later, we changed to a medium with iron limitation (CAA) without toxic components.

Additionally, regarding the concerns about the usage of Bip, we have now included Figure S6, which also refers to Query 18 of reviewer 1. Indeed, Bip was shown to be toxic by chelating other ions inside the cell [Henríquez, *et al.*, (2020), *Front Microbiol.* **11**, 1974]. However, Fig. S6 shows that growth for both the pyoverdine non-producer 3E2 and the triple deletion mutant $\Delta pm\Delta parX$ in KB plus 1 mM can be restored by addition of FeCl₃.

Query 3

The title mentions ParXY, but Only the parX gene has been deleted. What is the impact of deleting *parY* gene or both on the phenotype?

Answer: We did not inactivate the *parY* gene. We assumed that the deletion of one essential component of a tripartite efflux system inactivates the entire system. In fact, it was previously shown that the periplasmic adapter protein has an essential function for tripartite efflux pumps [Ge, *et al.*, (2009), *J Bacteriol.* **191**, 4365-71]. To our knowledge, the component anchored in the outer membrane can vary, and interactions between the cytoplasmic membrane component and the periplasmic adaptor protein are specific.

Query 4

The parXY efflux pump works with TtgC, is the phenotype persist with *ttgC* deletion mutant. it would be interesting to know if the ParXY efflux pump binds TtgC in this context or another outer membrane protein.

Answer: Yes, that's an interesting question. To the best of our knowledge, TtgC is proposed as an interaction partner [Puja, *et al.*, (2020), *Environ Microbiol.* **22**, 5222-31]. However, in this initial work on the possible role of the ParXY system in pyoverdine secretion, we did not look at other interaction partners of ParXY.

Query 5

Lines 117 to 123. The sentences correspond to a repetition of the previous section (introduction). The authors should focus on the new data.

Answer: Except for the first sentence, we deleted the statements to minimize repetition.

Query 6

The section on colony morphology is long. Only the essential messages should be provided. The paragraph needs to be revised and made more concise. Furthermore, all data are subjective and require quantitative values, colony size, fluorescence need to be measured and quantified. ParX deletion and complementation have a similar impact in iron-rich and iron-poor environments (Figure 2). How do you explain this?

Answer: The paragraph on colony morphology was revised and shortened as described in our response to reviewer 1.

As for the question of the common features of pyoverdine production under iron-replete and iron-deficient conditions (Fig. 2), the following observations should be considered:

1. KB (like CAA) is a medium that supports pyoverdine production, although pyoverdine is not necessary due to the high iron content of KB (about 7 μM in our case) [Zhang & Rainey 2013, *Evolution* 67, 3161].

2. The cells within the colonies are in a structured environment where pyoverdine is not a common good, unlike the well-mixed liquid culture (Zhang & Rainey 2013, *Evolution* 67, 3161). It can be assumed that the availability of iron varies greatly with the location of the cells in the colony.

Considering points 1 and 2, it is not surprising that after 18 h of incubation, the colonies look almost identical in terms of fluorescence on KB and KB plus 1 mM Bip. Our own and other work shows that only the dynamics of pyoverdine production are changed by the addition of Bip (after all, a non-producer can no longer grow on KB plus 1 mM Bip). In other work, we have distinguished producers and non-producers based on the color of the colonies [e.g., Becker et al. (2018) *Sci. Rep.* 8, 4093]. However, this was only successful at an early stage of colony development. With longer incubation, no significant differences between these strains could be detected visually.

Fig. 2 suggests that the additional deletion of *parX* impairs the production and secretion of pyoverdine in both KB and KB plus 1 mM Bip, an observation that is in good agreement with the other results presented in the manuscript. In addition, Fig. 2 shows that colony development of all three strains is similar on the KB plate, whereas the strain without *parX* grows poorly on the KB plus 1 mM Bip plate. This observation supports our hypothesis that *parX* affects pyoverdine synthesis and secretion in the absence of the two previously described systems.

Query 7

section "importance of *parX* for growth in liquid culture".

Lines 214 to 217. Results already presented on lines 140 to 145. Figures 3A and 3B correspond to the data shown in Figures 1 A and 1 B. How can you explain the difference in results for the 5-hour reading between the two figures?

Answer: Figure 1 shows the results of the screening and an abbreviated presentation of the results (only one growth parameter). Figure 3, on the other hand, shows the entire growth curves and includes additional strains for comparison. In addition, unlike Figure 1, growth is also measured in CAA medium. We would, therefore, leave the figure as it is. However, the manuscript text for Figure 3 has been shortened to avoid repetition.

Query 8

Lines 214 to 230: There is no new data compared with the results already presented.

Answer: We shortened the text to minimize repetition. Additionally, we also removed part of the supplementary material referred to in this paragraph.

Query 9

How do you explain the different results between the agar medium (impact of ParXY in the presence of high and low iron concentrations) and the liquid medium (impact of ParXY only in the low iron concentration)?

Answer: While nutrient conditions are largely identical for all cells in the well-mixed liquid culture, the nutrient supply for the cells in a colony is very heterogeneous. Cells at the edge of the colony find good nutrient conditions and presumably sufficient iron, while cells inside the colony are subject to nutrient limitation and iron deficiency. If pyoverdine is synthesized and secreted, the problem of iron deficiency in the colonies may be reduced. However, if there are limitations in these processes, a large proportion of cells in a colony may experience growth inhibition. In the case of a homogeneous distribution of cells and nutrients, including iron, this problem is not as pronounced. This is at least our explanation for the differences.

Query 10

Line 244. You suggest that *parX* complementation with the low copy number is responsible for the partial restoration of growth in CAA. But this is not the case in KB + BIP, and the same plasmid is used.

Answer: Expression of *parX* from plasmid pSEVA fully restores the growth rate in the exponential phase as well as the maximum growth yield in both CAA and KB + BIP media. Unlike KB + BIP, the lag phase remains prolonged with CAA medium (Fig. 4). The molecular basis for the latter difference is not known. In general, perfect complementation requires coordinated expression of the target gene in time and quantity. This can lead to problems in a plasmid-based complementation system with an IPTG-inducible promoter. Depending on the nutrient conditions (KB, rich medium; CAA, nutrient-poor medium), the problems may be phenotypically insignificant or noticeable (*e.g.*, prolonged lag phase in CAA medium). Based on these thoughts, we modified our statement in line 233 as follows:

“The prolonged lag phase in CAA medium, which is more nutrient-poor compared with KB, could be due to a temporal and/or quantitative imbalance of *parX* expression by the plasmid-based system.”

Query 11

Line 289. This is not very clear. According to Figure 7, *parX* expression already seems to be activated in iron-rich medium (at a basal level?). CAA medium, which contains more iron than KB + 1 mM BIP medium, seems to induce *parX* gene expression to a greater extent. Does this not agree with your hypothesis that *parX* is induced under low iron concentrations? The results should be checked by RT-qPCR. In addition, the expression level of the two-component system downstream of the *parX* gene should also be determined.

Answer: If *parXY* is indeed important for the secretion of pyoverdine, it is, at first sight, surprising that the expression of the gene is higher in CAA than in KB plus Bip. Unfortunately, we can only speculate about the cause of this difference. KB plus 1 mM Bip contains more iron (about 7 μ M), but this is bound in the Bip complex. CAA contains much less iron (about 1.5 μ M), which may be soluble or bound to proteins. Whether these parameters or other iron-independent differences of the medium are responsible for the different expression levels of *parXY* remains open. Therefore, in revising the manuscript, we concluded that no convincing conclusions about the importance of *parX* for iron acquisition can be drawn from the comparison of *parXY* expression levels in KB and CAA medium.

Consequently, we performed a new experiment showing that the expression of *parXY* in CAA medium is decreased by the addition of iron (Fig. 7). The section has been revised accordingly and now reads as follows (lines 289 – 296):

“Based on the high luminescence values measured compared with the negative control (reporter plasmid lacking the *parXY* promoter region), the *parXY* operon was induced in all media used. However, there were significant differences between the luminescence values that correlated with iron availability. For example, the addition of 10 μM FeCl_3 to the CAA medium resulted in an approximately 7-fold reduction in luminescence. Addition of 1 mM Bip to the KB medium resulted in a 2-fold increase in luminescence compared to KB alone. The results suggest that iron limitation stimulates the expression of the *parXY* operon”

Unfortunately, performing additional RT-qPCR experiments is beyond our possibilities at the moment.

Query 12

Supplementary Table S4 and Figures S1, S2, S3, S4, S5, S6 and S7 are not useful.

Answer: Supplementary Table S4 and Figures S3 and S4 were removed as suggested by the reviewer. Figure 2 was shortened. Figures S5 (now Fig. S4), S6 (now Fig. S5), and S7 (now Fig. S2) were kept in Supplementary Information. In our opinion, the graphs are not essential, but they confirm and complement the statements of the main text.

Once again, we would like to thank the reviewers for their constructive comments. We hope that the revisions made based on their feedback have improved the scientific rigor and clarity of our findings. We look forward to learning whether the changes made will enable the publication of the revised manuscript in *Microbiology Spectrum*.

Sincerely,
Heinrich Jung

August 28, 2023

Prof. Heinrich Jung
Ludwig Maximilian University of Munich
Planegg-Martinsried
Germany

Re: Spectrum02300-23R1 (The RND efflux system ParXY affects pyoverdine secretion in *Pseudomonas putida* KT2440)

Dear Prof. Heinrich Jung:

I appreciate your effort to amend the manuscript based on the Reviewers' comments. I am sincerely grateful to the Reviewers who did a great job in helping improve this manuscript with precious suggestions/comments. I think that the manuscript is now ready for publication. Hence, I am glad to communicate that your manuscript has been accepted, and I am forwarding it to the ASM Journals Department for publication. You will be notified when your proofs are ready to be viewed.

Sincerely,

Giordano Rampioni
Editor, Microbiology Spectrum
